# Redox Imbalance in Inflammation: The Interplay of Oxidative and Reductive Stress

**DOI:** 10.3390/antiox14060656

**Published:** 2025-05-29

**Authors:** Francesco Bellanti, Anna Rita Daniela Coda, Maria Incoronata Trecca, Aurelio Lo Buglio, Gaetano Serviddio, Gianluigi Vendemiale

**Affiliations:** C.R.E.A.T.E.—Center for Research and Innovation in Medicine, Department of Medical and Surgical Sciences, University of Foggia, 71122 Foggia, Italy; francesco.bellanti@unifg.it (F.B.); daniela.coda@unifg.it (A.R.D.C.); incoronata.trecca@unifg.it (M.I.T.); aurelio.lobuglio@unifg.it (A.L.B.); gaetano.serviddio@unifg.it (G.S.)

**Keywords:** redox imbalance, oxidative stress, reductive stress, inflammation, immune regulation

## Abstract

Redox imbalance plays a pivotal role in the regulation of inflammation, influencing both the onset and progression of various inflammatory conditions. While the pro-inflammatory role of oxidative stress (OS) is well established, the impact of reductive stress (RS)—a condition marked by excessive reducing equivalents such as NADH, NADPH, and reduced glutathione (GSH)—remains underappreciated. This review offers a novel integrative perspective by analyzing how OS and RS act not merely in opposition, but as interconnected modulators of immune function. We explore the mechanisms through which OS activates inflammatory pathways, and how RS, when sustained, can paradoxically impair immune defense, alter redox-sensitive signaling, and contribute to disease progression. Emphasis is placed on the dynamic interplay between these redox extremes and their combined contribution to the pathogenesis of chronic inflammatory diseases, including autoimmune, cardiovascular, and neuroinflammatory disorders. Additionally, we evaluate therapeutic strategies that target redox homeostasis, arguing for a shift from antioxidant-centric treatments to approaches that consider the bidirectional nature of redox dysregulation. This framework may inform the development of more precise interventions for inflammation-related diseases.

## 1. Introduction

Inflammation is a complex biological response to harmful stimuli, such as pathogens, damaged cells, or irritants, and is essential for maintaining tissue homeostasis and initiating repair mechanisms. Central to the regulation of inflammatory processes is the concept of redox balance—the equilibrium between oxidative and reductive forces within the cellular environment. Traditionally, research has predominantly focused on oxidative stress (OS), characterized by an overproduction of reactive oxygen (ROS) and nitrogen species (RNS), and its detrimental effects on inflammation and disease pathogenesis. However, emerging evidence underscores the equally significant, yet often overlooked, role of reductive stress (RS)—a condition where an excess of reducing equivalents disrupts redox homeostasis—in modulating inflammatory responses and contributing to various pathologies.

OS occurs when there is an imbalance favoring pro-oxidant species over antioxidant defenses, leading to potential damage to cellular components such as lipids, proteins, and DNA [1]. This state has been implicated in the initiation and progression of numerous inflammatory diseases, including cardiovascular disorders, diabetes, cancer, and neurodegenerative conditions [2]. ROS can activate redox-sensitive transcription factors like nuclear factor-kappa B (NF-κB), which in turn upregulates the expression of pro-inflammatory cytokines, perpetuating the inflammatory cascade [3].

Conversely, RS arises from an overabundance of reducing agents, such as NADH, NADPH, and glutathione, leading to an excessively reduced cellular state. In particular, RS is characterized by elevated NADH/NAD^+^ and NADPH/NADP^+^ ratios, an increased reduced/oxidized glutathione (GSH/GSSG) ratio, and the persistent activation of antioxidant systems. This imbalance can impair the formation of disulfide bonds in proteins, alter cellular signaling pathways, and compromise mitochondrial function [4]. Notably, chronic RS has been associated with the pathophysiology of various inflammation-related diseases, including certain cardiomyopathies, neurodegenerative disorders, and metabolic syndromes. For instance, excessive reducing equivalents can modulate the activity of NF-κB, thereby influencing inflammatory gene expression and contributing to disease progression [4].

Previous reviews have primarily focused on the pro-inflammatory effects of OS, or more recently, on the emerging role of RS in specific pathological contexts [4,5,6]. While each condition has been reviewed independently, an integrated perspective on their reciprocal regulation and combined influence on inflammation remains limited. This review addressed this gap by offering a critical synthesis of how OS and RS dynamically interact, how their dysregulation contributes to immune dysfunction and chronic disease progression, and why both extremes of the redox spectrum must be considered in therapeutic design. In doing so, we aim to move beyond descriptive summaries by identifying key conceptual and translational gaps in the current literature. Specifically, we highlight areas that require further investigation—such as the thresholds for harmful reductive stress, the dual role of Nrf2 in inflammation, and the risk of reductive imbalance induced by excessive antioxidant intake. We also discuss implications for future research, including the need for disease-specific redox profiling and biomarker development.

This review therefore provides a comprehensive and critical analysis of the dual facets of redox imbalance—OS and RS—and their respective roles in inflammation. By elucidating the molecular mechanisms underlying redox regulation of inflammatory processes, we seek to enhance the understanding of how deviations from redox homeostasis contribute to disease pathogenesis. Furthermore, we will explore potential therapeutic strategies targeting redox balance, offering insights into novel and more precise interventions for inflammation-associated diseases.

## 2. Oxidative Stress in Inflammation

OS represents a key pathogenic element in the physiopathology of chronic inflammatory diseases. This redox imbalance disrupts cellular signaling, biomolecular functionality, and structural integrity, leading to significant functional alterations. The relationship between OS and inflammation is not merely sequential but synergistic, establishing a pathogenic loop that sustains chronic inflammation. These dynamics have been extensively documented in several pathological conditions, including atherosclerosis, neurodegeneration, non-alcoholic steatohepatitis, inflammatory bowel diseases, and rheumatoid arthritis [2,7].

Further evidence from both clinical and experimental models has confirmed that OS not only exacerbates pre-existing inflammation but also plays a pivotal initiating role in the transition from acute to chronic inflammation. Elevated ROS levels influence the redox-sensitive activity of transcription factors such as NF-κB, the activating protein-1 (AP-1), and the hypoxia-inducible factor-1α (HIF-1α), which drive the expression of adhesion molecules, cytokines, chemokines, and matrix metalloproteinases (MMPs), thereby amplifying the tissue recruitment of immune cells and sustaining inflammatory circuits [8]. ROS also modulate inflammasome assembly, cellular metabolism, and epigenetic markers, which contribute to tissue remodeling, fibrosis, and loss of immune tolerance [9]. In chronic diseases such as metabolic syndrome and neuroinflammation, redox imbalance sustains a low-grade systemic inflammatory state (inflammaging) that underpins progressive tissue damage and functional decline [2].

### 2.1. Sources of Reactive Species

ROS and RNS are generated through a wide variety of intracellular mechanisms and play a dual role as both signaling mediators and damaging agents. Through the process of oxidative phosphorylation, mitochondria represent the primary source of endogenous ROS—which include superoxide anions (O_2_^•−^), hydrogen peroxide (H_2_O_2_), and hydroxyl radicals (^•^OH) produced through the Fenton reaction [10,11]. Although H_2_O_2_ is less reactive than O_2_^•−^, it is membrane-permeable and can propagate oxidative signaling to distant cellular compartments [12]. ^•^OH are among the most reactive and cytotoxic ROS, highly reactive species targeting DNA, lipids, and proteins [13].

Beyond mitochondria, additional cellular enzymes contribute to ROS generation. These include uncoupled nitric oxide synthase (NOS) isoforms, the cyclooxygenase (COX) family, NADPH oxidases (NOX1–5, DUOX1/2), and xanthine oxidase. NOX2, in particular, plays a central role in the oxidative burst essential for microbial killing in neutrophils and macrophages [14,15]. In addition, different isoforms of superoxide dismutase (SOD)—mitochondrial (SOD2), cytosolic (SOD1), and extracellular (SOD3)—mediate the dismutation of O_2_^•−^ into H_2_O_2_ and contribute to extracellular redox regulation [16]. Myeloperoxidase (MPO) further amplifies ROS production by converting H_2_O_2_ into hypochlorous acid in activated phagocytes [17].

On the nitrogen species side, inducible nitric oxide synthase (iNOS), strongly induced by pro-inflammatory cytokines, catalyzes the production of nitric oxide (NO^•^), which reacts rapidly with O_2_^•−^ to produce peroxynitrite (ONOO⁻), a powerful oxidant capable of nitrating tyrosine residues and damaging lipids, proteins, and DNA [18].

The endoplasmic reticulum (ER) also contributes to RS production during oxidative protein folding, particularly under conditions of ER stress. The activation of the unfolded protein response (UPR) leads to the increased expression of enzymes such as ER oxidoreductin 1 (Ero1) and protein disulfide isomerase (PDI), promoting disulfide bond formation and H_2_O_2_ release [19]. Chronic ER stress, often seen in metabolic diseases, links oxidative misfolding to inflammation via protein kinase R-like endoplasmic reticulum kinase (PERK), activating transcription factor 6 (ATF6), and inositol-requiring enzyme 1α (IRE1α) signaling branches [20]. Environmental factors, including ultraviolet (UV) radiation, tobacco smoke, heavy metals, hyperglycemia, and certain infections, can further potentiate ROS and RNS production through both mitochondrial and non-mitochondrial pathways, exacerbating redox imbalance and sustaining inflammation [21].

### 2.2. Oxidative Stress-Induced Signaling Pathways

NF-κB, a master regulator of the inflammatory response, represents the most well-characterized redox-dependent pathway [22]. NF-κB dimers are physiologically retained in the cytoplasm by IκB proteins, which can undergo phosphorylation and proteasomal degradation in response to OS and pro-inflammatory stimuli by activated IκB kinase (IKK) [23]. This event frees NF-κB to translocate into the nucleus, where it promotes the transcription of pro-inflammatory genes encoding cytokines, adhesion molecules, and enzymes like COX-2 and iNOS (Figure 1) [24].

In addition to NF-κB, mitogen-activated protein kinases (MAPKs), including ERK1/2, JNK, and p38 MAPK, are also activated in a redox-sensitive manner [25]. ROS inhibit MAPK phosphatases by oxidizing their catalytic cysteine residues, thereby prolonging MAPK signaling [26]. The p38 MAPK branch plays a key role in stabilizing mRNAs of inflammatory mediators and modulating chromatin accessibility via histone-modifying enzymes [27].

On the other hand, the nuclear factor erythroid 2-related factor 2 (Nrf2)–Keap1 pathway functions as a protective axis. Nrf2 is normally bound to Keap1, which facilitates its ubiquitination and degradation, but OS modifies critical cysteine residues on Keap1, leading to the stabilization and nuclear translocation of Nrf2 [28,29]. Once in the nucleus, Nrf2 binds to antioxidant response elements (AREs) and induces the transcription of cytoprotective genes such as heme oxygenase-1 (HO-1), NAD(P)H quinone dehydrogenase 1 (NQO1), and glutamate–cysteine ligase catalytic subunit (GCLC) [30]. Nrf2 activity not only mitigates oxidative damage but also exerts anti-inflammatory effects by modulating macrophage polarization and suppressing NF-κB-driven transcription [31].

### 2.3. Impact on Immune System and Cytokine Production

OS significantly affects both innate and adaptive immune responses. In innate immunity, ROS modulate the activation of various immune cells, particularly macrophages and neutrophils [32]. ROS are essential for the activation of the NOD-like receptor family pyrin domain containing 3 inflammasome, a multiprotein complex responsible for the cleavage of pro-caspase-1 into its active form. This in turn activates interleukin (IL)-1β and IL-18, key cytokines involved in the propagation of the inflammatory cascade [33].

Persistent ROS production can lead to the overactivation of the inflammasome, resulting in sterile inflammation, as observed in diseases such as gout, type 2 diabetes, and atherosclerosis [34]. In macrophages, OS influences polarization toward a pro-inflammatory M1 phenotype [35]. In dendritic cells, ROS enhance antigen presentation by upregulating the major histocompatibility complex (MHC)-II and co-stimulatory molecules (CD80 and CD86), thus promoting T cell activation [36].

In adaptive immunity, ROS levels influence T cell differentiation: moderate oxidative signals support Th1 and Th17 polarization, while high ROS suppress regulatory T cell (Treg) expansion by reducing forkhead box P3 expression [37].

This imbalance contributes to the maintenance of chronic inflammatory states and loss of immune tolerance. Additionally, OS can alter B cell maturation, antibody class switching, and antigen specificity [38]. RNS such as ONOO⁻ can further interfere with cytokine receptor signaling, for instance by nitrating tyrosine residues in IL-10 and tumor growth factor-β (TGF-β) receptors [39].

### 2.4. Role in Tissue Damage and Disease Progression

Chronic OS leads to cumulative and progressive cellular damage that contributes to the initiation and exacerbation of a wide range of pathological conditions. Lipid peroxidation, driven by ^•^OH and ONOO^−^, compromises membrane integrity and fluidity, and produces cytotoxic aldehydes such as malondialdehyde (MDA) and 4-hydroxynonenal (4-HNE). These aldehydes form covalent adducts with proteins and nucleic acids, impairing their structure and function and triggering inflammatory and autoimmune responses [39,40]. Protein oxidation results in the formation of carbonyl groups, disulfide cross-links, and nitrosylated residues, leading to enzyme inactivation and proteostasis disruption [41]. DNA oxidation includes base modifications such as 8-oxo-dG, single- and double-strand breaks, and DNA–protein crosslinks, promoting mutagenesis and increasing cancer risk [42].

At the tissue level, OS induces fibrotic remodeling through the activation of fibroblasts and upregulation of TGF-β signaling [43]. In the liver, this leads to the progression from steatosis to non-alcoholic steatohepatitis and fibrosis [44]. In the gut, OS disrupts epithelial tight junctions and promotes microbial translocation, fueling mucosal inflammation in inflammatory bowel disease (IBD) [45]. In the cardiovascular system, endothelial dysfunction driven by ROS is an early step in atherogenesis [46]. Systemically, OS perturbs insulin signaling and contributes to insulin resistance, dyslipidemia, and metabolic syndrome [47]. Furthermore, ROS and RNS play a role in aging and degenerative diseases, contributing to inflammaging and chronic low-grade inflammation [48].

## 3. Reductive Stress in Inflammation

### 3.1. Definition and Mechanisms of Reductive Stress

Cellular homeostasis depends on maintaining redox equilibrium. Redox reactions play a crucial role in various cellular processes, collectively referred to as redox signaling and redox regulation. Over the years, the field of redox biology has grown significantly, establishing itself as a key area of research. The concept of OS has also evolved to incorporate the role of redox signaling, prompting ongoing discussions on refining its definition [49,50]. OS has a dual role in biological systems: while excessive oxidants can damage biomolecules, moderate levels—termed oxidative eustress—are essential for redox signaling in normal physiological functions. Ongoing research investigates how redox signaling sustains this delicate balance through prevention, interception, and repair mechanisms, all regulated by specific pathways [51].

Another condition that disrupts redox balance is RS, which arises from an excess of reducing equivalents. This often results from impaired oxidative phosphorylation or prolonged activation of antioxidant pathways [52,53]. Importantly, the excessive intake of antioxidant-rich diets or supplements—especially in individuals without oxidative stress conditions—has also been implicated as a trigger for RS. Recent findings demonstrated that the chronic consumption of *Hibiscus sabdariffa* extract, a polyphenol-rich plant with strong antioxidant capacity, led to increased blood pressure, vascular dysfunction, and inflammatory responses in healthy animals, suggesting that the overstimulation of endogenous antioxidant systems can disrupt redox homeostasis [54,55]. If not properly regulated, RS can disrupt cellular signaling, impair cell differentiation, and contribute to the development of diseases such as cancer, diabetes, and cardiomyopathy [56].

To counteract RS, cells ubiquitinate the mitochondrial regulator folliculin-interacting protein 1 (FNIP1), targeting it for degradation. However, the mechanism by which the E3 ligase CUL2^FEM1B^ identifies FNIP1 based on its redox status—and how this process adapts to fluctuating cellular conditions—remains unclear [57,58]. CUL2^FEM1B^ has been shown to use zinc as a molecular scaffold to selectively recruit the reduced form of FNIP1 during RS. The ubiquitylation of FNIP1 is regulated by pseudosubstrate inhibitors from the brain-expressed X-linked (BEX) family, which prevent its premature degradation and protect cells from excessive ROS accumulation. Notably, a FEM1B gain-of-function mutation and BEX deletion lead to similar developmental disorders, underscoring the need for precise regulation of this zinc-dependent RS response to maintain both cellular and organismal balance [56].

While redox imbalance has traditionally been linked to excessive OS, growing evidence suggests that an overly reductive cellular environment can also drive cancer progression. RS occurs when antioxidant levels—such as glutathione (γ-l-glutamyl-l-cysteinylglycine) and NADH—become disproportionately high relative to oxidized NAD, disrupting key metabolic pathways essential for cell proliferation. In certain genetically driven cancers, targeting pathways involved in RS may present a promising therapeutic strategy. As redox-regulated mechanisms are increasingly recognized as critical control hubs, a deeper understanding of RS signaling could not only refine our knowledge of metabolic balance but also unveil new opportunities for cancer treatment [59].

Certain bacteria, including *Mycobacterium tuberculosis* (Mtb), encounter RS during infection. Host defense mechanisms—such as the acidic environment of phagosomes and lysosomes, glutathione exposure in alveolar lining fluid, and hypoxic conditions within granulomas—contribute to the accumulation of reduced cofactors like NADH, NADPH, FADH_2_, and nonprotein thiols [60]. This buildup induces RS in Mtb, which must be mitigated for survival. If left unchecked, RS can lead to the formation of reactive oxygen species, which can be lethal to the bacterium. In response, Mtb exhibits inhibited growth, metabolic shifts, altered virulence, and increased drug tolerance. To maintain redox homeostasis, the bacterium relies on thiol-based buffering systems, including mycothiol and ergothioneine, which function alongside the thioredoxin–thioredoxin reductase (TR) system. Additionally, Mtb modifies its tricarboxylic acid (TCA) cycle and dehydrogenase activity to dissipate RS. During prolonged exposure, it shifts to synthesizing storage and virulence lipids as an alternative redox management strategy [61]. Genetic screening could help identify key regulators of these pathways. Since both actively replicating and persistent Mtb cells depend on effective RS management, a deeper understanding of these mechanisms may reveal critical vulnerabilities for therapeutic intervention [62].

### 3.2. Antioxidant Systems

Cells have evolved molecular strategies to regulate ROS levels and minimize damage to macromolecules. Some mechanisms control the activity of enzymes responsible for ROS production, while others enhance ROS elimination or facilitate the repair of oxidative damage at affected sites [63].

In professional phagocytes, the NADPH oxidase complex transfers electrons from cytosolic NADPH to the phagocytic vacuole, generating O_2_^•−^. This process depends on the activation of flavocytochrome *b* by four cytoplasmic proteins. Its antimicrobial action involves K^+^ influx through BKCa channels, which raises vacuolar pH and activates neutral proteases. Similar enzyme systems are present in plants, lower animals, and humans, though their precise roles remain incompletely understood [64]. In mammals, a key antioxidant defense mechanism is the Nrf2/Keap1 pathway. This system is crucial for counteracting xenobiotic metabolism and ensuring a robust antioxidant response. By maintaining cellular redox homeostasis, it has played a significant role in the evolutionary adaptation of animals to changing environments. Within this regulatory network, Nrf2 acts as a transcription factor that regulates the expression of hundreds of genes, many of which help reduce ROS levels, while Keap1 functions as an intrinsic inhibitor of Nrf2 [65,66]. Glutathione is a tripeptide synthesized in the cytosol through two ATP-dependent steps catalyzed by γ-glutamylcysteine ligase and glutathione synthetase. Under normal conditions, most intracellular glutathione exists in its reduced form (GSH), with the oxidized form (GSSG) comprising less than 2%. GSH functions as an antioxidant by donating electrons via its cysteine thiol group to counteract ROS. During OS, glutathione peroxidase (GPX) reduces peroxides like H_2_O_2_ by oxidizing GSH to GSSG. Glutathione reductase then recycles GSSG back to GSH using NADPH. The GSH/GSSG ratio is crucial for maintaining redox balance [67].

Disruptions in glutathione balance can induce RS. Attempts to increase GSH levels—whether genetically or pharmacologically—have been associated with mitochondrial oxidation and cytotoxic effects [68]. Under excessively reduced conditions, mitochondria produce ROS at a rate that surpasses the cell’s antioxidant capacity, leading to oxidative damage despite the full activation of defense mechanisms [69].

Alongside Nrf2-mediated antioxidant elevation—marked by an increased GSH/GSSG ratio—elevated NADH levels play a significant role in the onset of RS [4]. NAD^+^ serves as a crucial electron carrier in redox reactions, transferring electrons from metabolic substrates to the mitochondrial electron transport chain, where it is oxidized to facilitate the conversion of oxygen into water. This electron flow drives oxidative phosphorylation, which is essential for ATP production. In pathological conditions, high blood glucose levels appear to be a major trigger of RS [70].

Among antioxidants, O_2_^•−^ plays a crucial role. It is first converted into H_2_O_2_ by SODs and then further broken down into water by catalase or peroxidases such as GPX. SODs exist in multiple forms—cytoplasmic, extracellular, and mitochondrial—each characterized by distinct structures, metal cofactors (e.g., Cu/Zn or Mn), and kinetic properties. Catalase, confined to peroxisomes, primarily acts on H_2_O_2_ at high concentrations due to its low substrate affinity. In contrast, GPX, present in both the cytosol and mitochondria, efficiently reduces H_2_O_2_ even at low levels by utilizing GSH [71]. GPX belongs to the thioredoxin superfamily of antioxidant enzymes, which includes thioredoxins (Trx), thioredoxin reductase (TrxR), glutaredoxins, and peroxiredoxins [72]. While Trx and glutaredoxin primarily reduce disulfide bonds, Trx, TrxR, and peroxiredoxins also play a direct role in detoxifying H_2_O_2_ [73]. Like SOD, these enzymes exist in multiple isoforms with distinct subcellular localizations, enabling ROS attenuation at various sites and concentrations within the cell. Many of these enzymes are inducible in response to OS, emphasizing the importance of tightly regulated ROS control in cellular homeostasis [74].

### 3.3. Effects of Excessive Reductive Conditions on Immune Function

Excessive reductive conditions can significantly impact immune function by altering redox-sensitive signaling pathways, impairing pathogen defense mechanisms, and contributing to chronic inflammation. RS arises when an imbalance in reducing equivalents, such as NADH, NADPH, and glutathione, leads to an overly reduced cellular state. While antioxidant mechanisms are crucial for mitigating oxidative damage, excessive reduction can interfere with redox-regulated immune signaling cascades and suppress effective immune responses (Figure 2) [75].

In innate immunity, phagocytes like macrophages and neutrophils rely on controlled oxidative burst to eliminate pathogens. A highly reduced intracellular status impairs NOX activity, suppressing O_2_^•−^ production and thus weakening microbial killing and overall host resistance [32]. In dendritic cells, excessive reducing conditions may alter antigen presentation by modulating the expression of major histocompatibility complex (MHC) molecules and costimulatory signals, potentially impairing T cell priming [76].

RS also significantly influences adaptive immunity. T cell receptor (TCR) signaling requires transient ROS generation for effective activation, proliferation and cytokine secretion [77]. Depletion of ROS under RS compromises these processes, leading to skewed T cell differentiation—favoring Treg over Th1 and Th17 subsets—and reducing immune competence [78]. B cell responses may also be impaired, as RS affects antibody production and class switching, as well as redox-sensitive transcription factors like B-cell lymphoma 6 (Bcl-6) and signal transducers and activators of transcription (STATs) involved in B cell fate decision [79].

At the transcriptional level, persistent RS alters the activity of key redox-sensitive transcription factors, including NF-κB and AP-1. While normally activated by moderate ROS levels to drive pro-inflammatory gene expression, these factors may be destabilized or improperly regulated in a reducing environment, resulting in aberrant or insufficient cytokine production [80]. In particular, reduced NF-κB activity has been linked to impaired TNF and IL-6 expression, potentially weakening acute immune responses and promoting low-grade chronic inflammation [80].

Beyond immunosuppression, RS contributes to chronic inflammatory diseases by disrupting metabolic and immune homeostasis. In metabolic disorders such as diabetes and obesity, high NADPH and GSH levels can impair insulin receptor signaling and support a pro-inflammatory state, contributing to the pathogenesis of metabolic inflammation [81]. Similarly, in neurodegenerative diseases, elevated antioxidant tone in glial cells may interfere with microglial ROS-dependent clearance mechanisms, exacerbating neuroinflammation, and neuronal damage [82].

Emerging evidence also suggests that excessive activation of the Nrf2 pathway under conditions of sustained RS can suppress innate immune signaling and interfere with pathogen recognition. While Nrf2 activation is protective against oxidative damage, its persistent stimulation may inhibit essential immune responses by downregulating TLR expression and type 1 interferon production, particularly during chronic inflammation or infection [83].

Overall, RS should not be viewed solely as the antithesis of OS, but rather as a distinct pathological condition that compromises immune integrity when unchecked. A tightly regulated redox environment is essential for orchestrating immune responses, and both oxidative and reductive extremes can lead to immune dysfunction. Future work is needed to better define the thresholds and biomarkers of RS and to develop targeted strategies that restore physiological redox signaling in inflammatory diseases.

### 3.4. Paradoxical Roles of Reductive Stress in Chronic Inflammation

Numerous studies have linked inflammation-associated RS to various pathologies. While traditionally viewed as protective, the excessive accumulation of reducing equivalents—such as NADPH and GSH—can paradoxically disrupt redox homeostasis, leading to cellular and tissue dysfunction. RS can hinder cell proliferation, interfere with disulfide bond formation during protein folding, impair mitochondrial oxidative phosphorylation, and suppress key signaling pathways involved in immune regulation and metabolism.

RS is increasingly implicated in a wide array of inflammation-related diseases, including protein aggregation cardiomyopathy, hypertrophic cardiomyopathy, muscular dystrophy, pulmonary hypertension, cancer, and autoimmune conditions such as rheumatoid arthritis [4]. These effects are often associated with characteristic shifts in redox biomarkers. An increased GSH/GSSG ratio is characteristic of conditions such as cardiomyopathy, stent stenosis, muscular dystrophy, and renal diseases, indicating an overly reduced intracellular environment that may compromise redox-sensitive regulatory systems. Similarly, an elevated NADPH/NADP^+^ ratio is reported in pulmonary hypertension, Parkinson’s disease, metabolic syndrome, rheumatoid arthritis, and various cancers, correlating with dysregulated redox signaling and altered cellular metabolism [65,84,85,86,87,88,89].

At the molecular level, Carne and colleagues demonstrated that RS in skin fibroblasts results in the selective downregulation of specific proteins, including key components of the MAPK signaling cascade and extracellular matrix elements [90]. Their findings suggest that excessive reducing agents can alter the phosphorylation patterns of signaling proteins, including enhanced Akt (protein kinase B) phosphorylation, potentially disrupting growth and survival pathways [90]. In cardiac tissue, sustained RS can promote the formation of misfolded and aggregated proteins due to inadequate disulfide bond formation, contributing to cardiomyopathies characterized by diastolic dysfunction and ventricular wall thickening [91].

In inflammatory conditions, a persistently reduced redox state may suppress necessary oxidative bursts required for immune cell activation, while simultaneously altering cytokine signaling. This paradox is evident in diseases such as rheumatoid arthritis, where both oxidative and RS biomarkers are elevated, reflecting a state of redox dysregulation rather than simple imbalance in one direction [92]. Furthermore, RS may contribute to the immune escape observed in some cancers, by modulating redox-sensitive checkpoints and impairing antigen presentation [93].

The dual and sometimes contradictory roles of RS highlight the complexity of redox regulation in chronic inflammation. It is not merely the presence of ROS or reducing agents, but their spatial, temporal, and quantitative control that determines the inflammatory outcome [94]. Recognizing RS as an active contributor to chronic inflammation opens new avenues for therapeutic intervention. Strategies aimed at restoring redox homeostasis—rather than simply scavenging ROS—are likely to offer more effective and nuanced approaches to managing inflammation-related diseases.

## 4. Interplay Between Oxidative and Reductive Stress

### 4.1. Dynamic Balance and Feedback Mechanisms

Redox homeostasis is essential for cellular function, immune signaling, and metabolic regulation. This balance is achieved through intricate feedback mechanisms that regulate the production and elimination of ROS/RNS. Disruptions in this equilibrium can lead to oxidative or RS, both of which are characterized by specific markers (Table 1) and have significant implications for inflammation and disease progression.

A key regulatory axis in redox homeostasis involves the redox-sensitive transcription factors NF-κB and Nrf2. Under OS conditions, NF-κB is activated, leading to the transcription of pro-inflammatory cytokines and enzymes that further elevate ROS levels, thereby amplifying the inflammatory response. Conversely, Nrf2 serves as a counter-regulatory factor by inducing the expression of antioxidant proteins that mitigate oxidative damage and restore redox balance. Upon OS, it escapes Keap1-mediated degradation, translocates to the nucleus, and induces genes involved in detoxification and redox buffering. The interplay between NF-κB and Nrf2 exemplifies a dynamic feedback loop in which inflammatory signaling enhances OS, which in turn activates antioxidant responses aimed at re-establishing equilibrium [99].

Enzymatic systems reinforce this feedback structure. NOXs produce ROS in a regulated fashion, particularly in immune cells for antimicrobial defense and redox signaling. Their isoforms (NOX1-5 and DUOX1/2) exhibit tissue-specific expression and regulation, contributing to both physiological signaling and pathological ROS overproduction [100]. SODs convert O_2_^•−^ to H_2_O_2_, which is subsequently reduced to water by catalase and GPX. SOD enzymes exist in mitochondrial, cytosolic, and extracellular isoforms and play critical roles in limiting oxidative damage and shaping redox signaling [101]. The Trx and glutaredoxin systems also maintain redox homeostasis by modulating protein thiol oxidation, thereby influencing various signaling pathways involved in inflammation. Dysregulation in any of these systems can tip the redox state toward either oxidative or reductive extremes [102].

Redox balance is further modulated by immunometabolic shifts. Activated immune cells undergo metabolic reprogramming that affects redox tone. For instance, itaconate, a metabolite produced in inflammatory macrophages, exerts anti-inflammatory effects by inhibiting succinate dehydrogenase and limiting mitochondrial ROS production. This redox–metabolic coupling serves as a feedback mechanism, where metabolic intermediates adjust redox signaling and immune function in tandem [103]. Beyond itaconate, other metabolites such as succinate and fumarate can stabilize HIF-1α or inhibit α-ketoglutarate-dependent dioxygenases, thereby influencing inflammatory gene expression. Conversely, ROS/RNS regulate metabolic pathways by reversibly modifying thiol groups on metabolic enzymes, impacting glycolysis, fatty acid oxidation, and mitochondrial respiration. This bidirectional interaction between redox signaling and metabolism is especially prominent in activated macrophages, T cells, and endothelial cells [7,104].

Disruptions in these feedback mechanisms—such as persistent NF-κB activation alongside insufficient Nrf2 activity—can lead to a state of chronic redox imbalance and unresolved inflammation. Thus, understanding the interplay between oxidative and reductive responses, and the feedback networks that regulate them, is critical for developing therapeutic strategies aimed at restoring redox balance and resolving inflammation.

### 4.2. Conditions Leading to Redox Imbalance

Redox homeostasis is essential for maintaining cellular integrity, metabolic regulation, and immune function. Multiple endogenous and exogenous factors can disrupt this equilibrium, tipping the balance toward either oxidative or RS and promoting inflammation and disease progression, as follows:Environmental and lifestyle factors: Exposure to pollutants, heavy metals, tobacco smoke, radiation, and various xenobiotics can elevate ROS/RNS production, overwhelming antioxidant defense mechanisms and leading to OS [103]. Additionally, poor dietary habits, sedentarism, and chronic psychological stress contribute to systemic redox imbalance and heighten susceptibility to inflammation-related diseases [105];Inflammatory processes: The activation of immune cells, such as macrophages and neutrophils, during infections or chronic inflammation leads to excessive production of ROS/RNS, amplifying OS and tissue damage. This persistent redox imbalance is observed in conditions such as rheumatoid arthritis and inflammatory bowel disease [106];Genetic mutations and hereditary diseases: Inherited diseases including cystic fibrosis, familial amyotrophic lateral sclerosis (ALS), and certain mitochondrial encephalopathies are associated with chronic redox imbalance. Mutations in genes regulating antioxidant enzymes, mitochondrial function, or protein folding exacerbate ROS production and trigger inflammation through UPR and mitochondrial damage [107];Metabolic dysregulation: Conditions such as obesity, insulin resistance, and type 1 and type 2 diabetes are associated with heightened OS. Excess adipose tissue and hyperglycemia stimulate mitochondrial ROS production and NOX activity, impairing redox signaling and promoting systemic inflammation [108]. Mitochondria isolated from the carotid body of type 1 diabetic rats exhibited evidence of RS, supporting the role of mitochondrial redox imbalance in diabetic pathology and systemic inflammation [109]. In metabolic syndrome, this imbalance contributes to endothelial dysfunction, vascular inflammation, and progression to cardiovascular disease (CVD) [110];Aging: The aging process is associated with a decline in mitochondrial efficiency and antioxidant capacity, resulting in cumulative oxidative damage and the emergence of a pro-inflammatory state known as “inflammaging” [111]. Redox imbalance during aging contributes to tissue degeneration, neuroinflammation, and increased risk of chronic diseases such as Alzheimer’s and atherosclerosis [112];Endothelial dysfunction: An imbalance between NO bioavailability and OS in the endothelium contributes to impaired vasodilation, vascular inflammation, and thrombosis. This mechanism plays a central role in the development of hypertension and atherosclerosis, linking redox imbalance to cardiovascular events [113].

Collectively, these diverse factors converge to disrupt redox homeostasis, highlighting the importance of personalized strategies to correct redox imbalance and mitigate inflammation-driven diseases.

### 4.3. Consequences of Redox Dysregulation in Chronic Inflammatory Diseases

The disruption of redox homeostasis is a central driver of chronic inflammatory diseases, influencing immune cell function, promoting tissue damage, and fueling metabolic dysfunction (Figure 3).

The sustained activation of NF-κB and AP-1 by persistent OS drives the overproduction of pro-inflammatory cytokines (e.g., TNF, IL-6, and IL-1β). This creates a self-perpetuating inflammatory loop that contributes to disease progression [2]. In diseases such as rheumatoid arthritis and inflammatory bowel disease, prolonged OS induces oxidative modifications to DNA, proteins, and lipids, which compromise cellular integrity. These changes impair mitochondrial function, induce genomic instability, and promote apoptotic or necrotic cell death, all of which exacerbate tissue injury and inflammation [114]. In neurodegenerative disorders such as Alzheimer’s and Parkinson’s disease, OS exacerbates neuroinflammation by impairing mitochondrial function and disrupting proteostasis, contributing to neuronal death and cognitive decline [115].

While traditionally underexplored, RS has also emerged as a contributor to chronic disease. Excess reducing equivalents—such as NADH, NADPH, and GSH—can interfere with redox-sensitive signaling pathways, required for immune activation, hinder disulfide bond formation, and stabilize misfolded proteins, all of which impair cellular function and immune surveillance [116]. In metabolic and cardiovascular diseases, this dual imbalance is particularly evident. On the one hand, OS drives endothelial dysfunction, insulin resistance, and atherosclerotic plaque formation; on the other, excessive RS may impair NO bioavailability and redox signaling, thereby exacerbating vascular stiffness and hypertensive pathology [117,118].

In chronic pulmonary and renal disorders, redox dysregulation promotes fibroblast activation and excessive extracellular matrix deposition, leading to tissue fibrosis and irreversible organ damage. This is mediated through redox-sensitive pro-fibrotic pathways, including TGF-β and HIF-1α, which are activated or dysregulated under conditions of oxidative and reductive imbalance [119].

Collectively, these findings highlight the crucial role of redox homeostasis in immune regulation, tissue repair, and metabolic control. Both oxidative and reductive extremes contribute to pathological inflammation, and therapeutic efforts should aim not merely to suppress ROS but to restore redox homeostasis as a dynamic and context-dependent physiological state.

## 5. Redox Imbalance in Chronic Inflammatory Diseases

Chronic inflammatory diseases are characterized by persistent immune activation, often driven by dysregulated redox signaling that disrupts cellular homeostasis. Redox imbalance, whether due to excessive OS or aberrant reductive conditions, alters immune cell function, promotes tissue injury, and exacerbates disease progression (Table 2).

Understanding how redox perturbations contribute to chronic inflammation is essential for identifying novel therapeutic strategies aimed at restoring balance and preventing long-term tissue damage [4,127].

### 5.1. Redox Perturbations in Autoimmune Disorders

Redox imbalance plays a pivotal role in the pathogenesis of autoimmune disorders such as rheumatoid arthritis (RA), systemic lupus erythematosus (SLE), type 1 diabetes mellitus (T1DM), autoimmune hepatitis (AIH), and autoimmune thyroiditis (AIT).

In RA, heightened OS leads to the oxidative modification of proteins and lipids within the synovial joints, contributing to inflammation and joint damage. Elevated levels of MDA, a marker of lipid peroxidation, have been observed in RA patients, indicating increased oxidative damage. Additionally, antioxidant enzymes including SOD and GPX are often altered in RA; these enzymes are typically overexpressed and show increased enzymatic activity as part of a compensatory antioxidant response to elevated OS, suggesting an overwhelmed antioxidant defense system [120,121].

Similarly, in SLE, OS contributes to disease pathology by promoting the formation of autoantigens through oxidative modifications of cellular components. These modified molecules can trigger autoantibody production, leading to immune complex deposition and subsequent tissue damage. Furthermore, OS in SLE has been linked to endothelial dysfunction, accelerating atherosclerosis and increasing cardiovascular risk in affected individuals [128].

T1DM involves autoimmune destruction of pancreatic β-cells, with OS playing a significant role in disease progression. The infiltration of macrophages and dendritic cells into pancreatic islets leads to the overproduction of ROS, which, coupled with compromised antioxidant defenses, contributes to β-cell apoptosis and impaired insulin secretion [129]. Additionally, the activation of the polyol pathway in hyperglycemic conditions exacerbates redox imbalance by increasing NADH levels and reducing NAD+ availability, further promoting OS and cellular damage [130].

In AIH, increased OS has been implicated in liver tissue damage and fibrosis. Studies have demonstrated elevated levels of lipid and protein oxidative injury products, such as 8-isoprostane, in both urine and plasma of AIH patients, indicating significant OS [131]. Additionally, decreased glutathione levels and altered antioxidant enzyme activities have been observed, suggesting an overwhelmed antioxidant defense system [132].

In AIT, including Hashimoto’s thyroiditis (HT) and Graves’ disease (GD), OS contributes to thyroid cell damage and the autoimmune response. Enhanced OS in HT patients has been associated with increased DNA damage and lipid peroxidation. Furthermore, OS may disrupt self-tolerance in the thyroid, leading to autoimmune thyroid dysfunction. In GD, oxidative DNA damage appears to play a significant role in its pathogenesis, with increased markers of OS correlating with disease severity [133].

The dual role of ROS in autoimmune and inflammatory diseases underscores the necessity of maintaining redox balance; while adequate ROS levels are essential for normal immune function, excessive ROS can lead to tissue damage and exacerbate disease progression [134]. Understanding the mechanisms by which redox perturbations influence autoimmune pathophysiology is crucial for developing targeted therapeutic strategies aimed at restoring redox homeostasis and mitigating inflammation in these disorders.

Recent findings also highlight the role of RS in shaping immune dysfunction across autoimmune disorders. In T1DM, as reported previously, reductive shifts in mitochondrial redox state—evidenced by elevated NADH/NAD^+^ ratios—have been observed in the carotid body, impairing metabolic sensing and contributing to sympathetic overactivation and systemic inflammation [109]. This suggests that reductive stress may coexist with oxidative damage, compounding immune dysregulation.

In SLE, increased levels of NADPH and GSH have been reported during certain disease stages, potentially reflecting compensatory responses to ROS. However, when sustained, these responses may suppress ROS-dependent signaling needed for proper immune resolution and antigen clearance [135]. This could explain the paradox of persistent inflammation in a biochemically reduced intracellular environment.

In autoimmune thyroid disorders, the overexpression of antioxidant enzymes such as SOD and catalase has been described in thyroid tissues, possibly contributing to an overly reduced local milieu [133]. Such conditions may impair redox-sensitive apoptotic pathways, allowing autoreactive immune cells to persist.

Autoimmune hepatitis (AIH) may similarly involve both oxidative and reductive mechanisms. While oxidative damage drives hepatocyte apoptosis and fibrogenesis, emerging data suggest that aberrant activation of Nrf2 and elevated GSH levels may contribute to immune evasion and chronic inflammation, supporting a dual-phase redox model in AIH progression [136].

These observations collectively reinforce the idea that autoimmune disorders are not defined by OS alone, but by a dysregulated redox spectrum—where both excessive oxidation and sustained reduction impair immune balance. Tailored interventions must consider disease stage and redox context to effectively restore immune homeostasis.

### 5.2. Redox Perturbations in Cardiovascular Diseases

Redox alterations are determinant in the pathogenesis of various CVDs through promotion of inflammatory processes. In conditions such as atherosclerosis, hypertension, and heart failure, elevated OS induces endothelial dysfunction by reducing NO bioavailability, a critical regulator of vascular tone and health. This reduction in NO leads to impaired vasodilation and heightened vascular resistance, fostering an environment conducive to inflammation. The resulting endothelial dysfunction is marked by increased expression of adhesion molecules and pro-inflammatory cytokines, facilitating the recruitment and infiltration of immune cells into the vascular wall. This cascade of events contributes to the initiation and progression of atherosclerotic lesions and other cardiovascular pathologies [137].

Furthermore, OS activates redox-sensitive transcription factors such as NF-κB, which upregulates the expression of inflammatory genes, amplifying the inflammatory response within the cardiovascular system. This chronic low-grade inflammation, driven by persistent redox imbalance, is a hallmark of many CVDs and represents a significant therapeutic target [138].

In heart failure, the interplay between redox imbalance and inflammation is particularly evident. Mitochondrial dysfunction leads to excessive ROS production, which not only damages cardiac myocytes but also activates inflammatory pathways, exacerbating myocardial injury and impairing cardiac function [139].

Emerging evidence indicates that OS also plays a critical role in ischemia-reperfusion injury, a major contributor to myocardial infarction and subsequent heart failure. During reperfusion, a burst of ROS leads to oxidative damage, mitochondrial permeability transition pore opening, and the activation of cell death pathways including necrosis and apoptosis [140]. Inflammatory cytokine production is rapidly triggered, promoting further damage to the myocardium and limiting recovery [141].

Beyond OS, emerging evidence highlights the detrimental effects of RS on cardiovascular health [142]. In the heart, RS has been linked to mitochondrial dysfunction, cardiomyopathy, and heart failure [143]. For instance, chronic RS has been shown to cause pathological cardiac remodeling and diastolic dysfunction in mice due to thiol–redox imbalance and the maladaptive overactivation of antioxidant pathways, including the persistent upregulation of enzymes such as catalase, (GPX), and thioredoxin reductase [91]. This overly reduced intracellular state interferes with redox signaling, calcium homeostasis, and contractile protein structure, thereby promoting fibrosis and impairing myocardial function [142].

Addressing redox perturbations in CVDs necessitates a nuanced approach that considers both oxidative and reductive imbalances. Therapeutic strategies aimed at restoring redox homeostasis, such as modulating antioxidant levels and targeting specific redox-sensitive signaling pathways, hold promise in mitigating inflammation-driven cardiovascular damage. However, the indiscriminate use of antioxidants may exacerbate RS, underscoring the importance of personalized interventions based on individual redox profiles.

In atherosclerosis, OS is not only a by-product of vascular inflammation but a primary driver of lipid oxidation, foam cell formation, and vascular smooth muscle cell (VSMC) proliferation. Oxidized LDLs (oxLDLs) promote endothelial activation and monocyte recruitment, while simultaneously activating NOX enzymes and uncoupled eNOS, further increasing ROS production [144]. This amplifies vascular injury and propagates plaque instability. In advanced lesions, both oxidative and reductive shifts can coexist, contributing to complex redox microenvironments within atherosclerotic plaques [145].

Hypertension, traditionally associated with enhanced vascular ROS production and impaired NO bioavailability, has also been linked to RS in experimental models. The increased activity of G6PD and elevated NADPH levels have been observed in hypertensive states, promoting enhanced superoxide generation via NOX enzymes [146]. Additionally, the excessive upregulation of Nrf2-target genes in the vasculature can impair ROS-dependent signaling necessary for vascular tone adaptation, contributing to maladaptive remodeling and stiffness [147].

In cardiac hypertrophy and diastolic heart failure, RS may be driven by the persistent overexpression of antioxidant enzymes (e.g., catalase, GPX, and thioredoxin reductase) in the absence of corresponding ROS elevation. This creates an excessively reduced intracellular environment that disrupts thiol–redox signaling, impairs calcium handling, and contributes to sarcomeric dysfunction and myocardial fibrosis [148].

Redox misregulation also affects autonomic control in CVDs. Both OS and RS in the central nervous system can impair baroreceptor reflexes and enhance sympathetic outflow, promoting neurogenic hypertension and systemic inflammation [149].

### 5.3. Loss of Redox Balance in Neuroinflammatory Conditions

Elevated ROS levels can damage neuronal lipids, proteins, and DNA, leading to neuronal dysfunction and death. This OS not only inflicts direct neuronal injury but also activates microglia and astrocytes, the primary immune cells of the central nervous system (CNS), thereby exacerbating neuroinflammation.

In Alzheimer’s disease (AD), OS contributes significantly to the aggregation of amyloid-beta (Aβ) plaques and hyperphosphorylated tau protein, both hallmark features of the disease. OS is known to play an important role in the pathogenesis of AD, with Aβ inducing OS and vice versa, creating a detrimental cycle [124]. Additionally, OS has been observed prior to Aβ plaque formation, suggesting its role in the early stages of AD pathology [150].

Parkinson’s disease (PD) is marked by the degeneration of dopaminergic neurons in the substantia nigra. Mitochondrial dysfunction and subsequent ROS overproduction are central to PD pathogenesis, resulting in oxidative damage to neuronal components and activation of neuroinflammatory pathways. The OS-induced activation of microglia leads to the release of pro-inflammatory cytokines, further contributing to neuronal damage [151].

In multiple sclerosis (MS), an autoimmune demyelinating disorder, OS exacerbates neuroinflammation and neuronal injury. Activated microglia and infiltrating macrophages produce ROS and RNS, which damage oligodendrocytes and myelin sheaths, impairing neuronal signal transmission [152]. This oxidative damage not only contributes to demyelination but also promotes axonal degeneration, leading to neurological deficits [153].

In neuroinflammatory conditions, RS can impair mitochondrial function, leading to decreased ATP production and increased ROS generation. This paradoxical increase in ROS under reductive conditions can further exacerbate oxidative damage, creating a vicious cycle of redox imbalance. Additionally, RS may promote protein misfolding and aggregation, contributing to the pathogenesis of neurodegenerative diseases [154].

Beyond the canonical neurodegenerative diseases, other neurological disorders also exhibit redox-linked pathology. In amyotrophic lateral sclerosis, both familial and sporadic forms show early mitochondrial dysfunction, increased protein oxidation, and glutathione depletion in spinal motor neurons. Redox dysregulation contributes to motor neuron degeneration through excitotoxicity and inflammation [155].

Huntington’s disease, a genetic neurodegenerative disorder, also demonstrates early oxidative stress in striatal neurons, with altered redox enzyme expression and evidence of lipid and DNA oxidation [156]. Antioxidant enzyme overexpression in transgenic models has delayed disease onset, yet sustained elevation of reducing equivalents may pose risks of RS in advanced disease stages [157].

While oxidative stress has long been a central focus in neurodegeneration, emerging data suggest that RS may contribute to neuroinflammation by suppressing essential redox signaling and promoting proteostasis disruption. For instance, the overactivation of the Nrf2 pathway in astrocytes has been associated with reductive stress in the aging brain, potentially impairing glial–neuronal communication [158]. Likewise, sustained high GSH levels in neuroblastoma cells under stress conditions have been shown to inhibit autophagy, an essential mechanism for clearing misfolded proteins and damaged mitochondria [159].

This dual imbalance may be especially relevant in aging-related neuroinflammation, where the redox buffering system becomes dysregulated. The overlap of oxidative damage and compensatory reductive shifts may explain why traditional antioxidant therapies often fail to improve clinical outcomes in AD and PD.

The intricate interplay between redox imbalance and neuroinflammation suggests that therapeutic strategies aimed at restoring redox homeostasis could mitigate neuroinflammatory damage. Antioxidant therapies, such as the use of polyphenols, have shown promise in reducing OS and modulating neuroinflammatory responses [160]. For instance, plant polyphenols have been reported to support longevity and have anti-aging properties that can slow brain aging, improve immune function, and protect against age-related diseases [161,162].

Understanding the mechanisms by which redox perturbations influence neuroinflammatory conditions is crucial for developing targeted interventions aimed at preserving neuronal function and preventing disease progression.

## 6. Therapeutic Strategies Targeting Redox Balance

The close link between redox disbalance and the pathogenesis of chronic inflammatory diseases highlights the need for therapeutic strategies aimed at restoring this equilibrium [163,164,165]. Beyond the use of antioxidants to counteract excessive ROS, emerging approaches include targeting redox-sensitive transcription factors to modulate inflammatory responses [166,167]. For instance, pharmacological agents that influence transcription factors such as NF-κB and Nrf2 have shown promise in reducing inflammation [168,169]. Additionally, lifestyle interventions, such as dietary modifications rich in bioactive compounds and regular physical activity, have been associated with improved redox balance and reduced inflammatory markers [170,171]. Collectively, these strategies offer promising avenues for the management of chronic inflammatory conditions (Table 3).

### 6.1. Antioxidant Therapies and Their Limitations

Antioxidant therapies have been extensively explored as potential treatments for chronic inflammatory diseases, aiming to attenuate ROS and mitigate OS. Despite the theoretical benefits, clinical outcomes of antioxidant interventions have been inconsistent, highlighting several limitations [177].

One significant challenge is the bioavailability and pharmacokinetics of antioxidant compounds. Many antioxidants exhibit poor absorption, rapid metabolism, and swift elimination, reducing their therapeutic efficacy [178]. For instance, curcumin, derived from turmeric, demonstrates potent anti-inflammatory properties in vitro but suffers from low systemic availability in vivo [179]. Additionally, the non-specificity of antioxidants poses concerns. While they aim to mitigate excess ROS, indiscriminate scavenging can disrupt essential redox signaling pathways vital for normal cellular functions. This lack of specificity may lead to unintended consequences, such as impairing immune responses or interfering with cellular signaling mechanisms [178]. Moreover, the timing and context of antioxidant administration are critical. In certain scenarios, particularly when tissue damage has already occurred, antioxidant therapy may be less effective or even detrimental. For example, in radiation-induced lung injury, once damage initiates, antioxidants often fail to halt the progression of tissue injury as other pathological factors become predominant [180]. Furthermore, the complexity of OS in disease pathology complicates therapeutic approaches. OS can be both a cause and a consequence of disease processes, making it challenging to determine when and how to intervene effectively. This dual role necessitates a nuanced understanding of disease-specific redox biology to tailor antioxidant therapies appropriately [178].

It is also essential to distinguish between stoichiometric antioxidants—compounds that directly scavenge ROS in a 1:1 reaction—and agents that modulate endogenous antioxidant pathways. Stoichiometric antioxidants, such as vitamins C and E, can cause reductive stress (RS) when used excessively or inappropriately.

Over-supplementation may shift the redox environment toward an excessively reduced state by increasing the levels of NADH, NADPH, and GSH, thereby impairing redox-sensitive signaling and contributing to immune dysfunction and metabolic imbalance [81]. In summary, while antioxidant therapies hold promise for managing chronic inflammatory diseases, their application is hindered by issues related to bioavailability, specificity, timing, and the intricate nature of OS in disease mechanisms. Moreover, future antioxidant therapies should aim for a balanced redox modulation rather than indiscriminate ROS scavenging. Addressing these challenges requires a comprehensive approach, integrating insights from redox biology, pharmacology, and clinical research to develop more effective and targeted antioxidant-based interventions.

### 6.2. Modulating Redox-Sensitive Signaling Pathways

Modulating redox-sensitive signaling pathways has emerged as a promising therapeutic strategy for managing chronic inflammatory diseases. These pathways, sensitive to the cellular redox state, play pivotal roles in regulating inflammation and immune responses.

Nrf2 is a key transcription factor that governs the expression of antioxidant and cytoprotective genes. The activation of Nrf2 enhances the cellular antioxidant capacity, thereby mitigating OS and its associated inflammatory responses. Conversely, NF-κB is a central regulator of pro-inflammatory gene expression. Under OS conditions, NF-κB activation leads to the production of various inflammatory mediators, perpetuating chronic inflammation. The interplay between Nrf2 and NF-κB is complex; for instance, Nrf2 can inhibit NF-κB activity, suggesting that enhancing Nrf2 function may concurrently suppress inflammatory pathways mediated by NF-κB [181,182,183].

Pharmacological agents and natural compounds that modulate these transcription factors have been investigated for their therapeutic potential. Polyphenols, such as curcumin and resveratrol, have demonstrated the ability to activate Nrf2 while inhibiting NF-κB, thereby exerting anti-inflammatory effects [184,185]. Additionally, certain phytochemicals can modulate other redox-sensitive transcription factors like AP-1 and signal transducer and activator of transcription 3 (STAT3), further influencing inflammatory responses [186]. Beyond transcription factors, other redox-sensitive signaling molecules, such as MAPKs and phosphoinositide 3-kinase/Akt pathways, are implicated in inflammation regulation [187,188]. Targeting these pathways with specific inhibitors or activators can modulate inflammatory responses, offering additional therapeutic avenues.

Among pharmacological Nrf2 activators, dimethyl fumarate (Tecfidera), monomethyl fumarate (Bafiertam), and diroximel fumarate (Vumerity) have been approved for the treatment of multiple sclerosis. These agents act via mild electrophilic stress to trigger Nrf2-dependent transcriptional programs, leading to the increased expression of antioxidant enzymes and suppression of inflammatory mediators [189,190,191]. Notably, Vumerity has demonstrated not only redox modulation but also anti-inflammatory effects, underscoring its relevance in redox-targeted therapy [192]. In addition to fumarate derivatives approved for multiple sclerosis, the Nrf2 activator omaveloxolone has recently received fast-track approval for the treatment of Friedreich’s ataxia [193]. Omaveloxolone enhances Nrf2 signaling via Keap1 inhibition and has shown efficacy in improving mitochondrial function and reducing oxidative damage in neurodegenerative contexts [194]. These compounds exemplify the therapeutic value of Nrf2 activation in diseases where both OS and inflammation are central to pathophysiology.

However, the therapeutic modulation of redox-sensitive pathways presents challenges. The ubiquitous nature of these pathways means that interventions can have widespread effects, necessitating precise targeting to avoid unintended consequences. Moreover, the dual roles of certain transcription factors in both promoting and resolving inflammation complicate therapeutic strategies. In addition, the excessive activation of the Nrf2 pathway can lead to pathological RS, suppressing immune defense mechanisms and promoting a tumor-friendly microenvironment [195]. The excessive intake of high-antioxidant foods, such as *Hibiscus sabdariffa*, can lead to increased Nrf2 activity, elevated antioxidant defenses, and a consequent reductive shift associated with inflammation and vascular dysfunction [54]. These findings further support the hypothesis that sustained Nrf2 upregulation may contribute to immune suppression and disease progression in specific contexts. Therefore, a nuanced understanding of the specific roles and interactions of these pathways in various disease contexts is essential for developing effective interventions.

### 6.3. Pharmacological and Dietary Approaches

Pharmacological and dietary interventions have been extensively investigated for their potential to restore redox balance and mitigate chronic inflammation. These approaches aim to modulate OS and inflammatory responses through various mechanisms, offering promising avenues for managing chronic inflammatory diseases.

Pharmacological strategies often involve the use of agents that can modulate OS and inflammatory pathways. For instance, certain medications have been observed to exhibit anti-inflammatory and immunomodulatory properties by decreasing the production of ROS and NO, thereby reducing OS and inflammation [123]. These effects are thought to be mediated through the inhibition of NF-κB. Such properties have been explored in conditions like inflammatory bowel disease and periodontitis [196,197]. Additionally, the use of iron chelators has been proposed as a therapeutic strategy. Inappropriate iron chelation can contribute to the production of ^•^OH, exacerbating chronic inflammation [198]. Effective chelation of iron by natural or synthetic ligands may help mitigate this oxidative damage, offering potential benefits in various inflammatory and degenerative diseases [199].

Dietary interventions play a crucial role in modulating redox balance and inflammation. Adherence to diets rich in antioxidants and anti-inflammatory components has been associated with reduced OS and improved clinical outcomes in chronic inflammatory conditions. For example, the Mediterranean diet, characterized by a high consumption of fruits, vegetables, whole grains, nuts, and olive oil, has been linked to lower levels of OS and inflammation. Studies have shown that individuals adhering to this diet exhibit improved biomarkers of redox balance and reduced incidence of chronic diseases such as cardiovascular disorders [200]. Similarly, plant-based diets have been associated with decreased OS and inflammation. A study indicated that such dietary patterns correlate with lower levels of biomarkers indicative of OS and inflammation, suggesting their potential in preventing and managing chronic diseases [201]. Incorporating specific functional foods and beverages can further enhance redox balance. For instance, regular consumption of green tea, rich in polyphenols like epigallocatechin gallate, has been shown to reduce inflammation and support cardiovascular health [202]. Moreover, the inclusion of spices such as turmeric, containing the active compound curcumin, in the diet has demonstrated anti-inflammatory and antioxidant effects. Combining turmeric with black pepper enhances curcumin absorption, amplifying its therapeutic potential [203]. It is important to note that while dietary antioxidants offer health benefits, their efficacy can be influenced by factors such as bioavailability and individual metabolic responses. Therefore, personalized dietary recommendations, possibly in consultation with healthcare professionals, can optimize the therapeutic outcomes of these interventions.

It is worth considering that, even though dietary and pharmacological interventions, including polyphenols, omega-3 fatty acids, and caloric restriction mimetics, have been proposed to restore redox balance, excessive metabolic shifts towards a reductive state (e.g., through prolonged fasting or high-dose NAD+ precursors) could impair mitochondrial function and lead to RS-associated diseases, such as neurological disorders and metabolic syndromes [204]. A precision nutrition approach is necessary to prevent an unintended reductive shift, particularly in patients prone to metabolic dysregulation.

### 6.4. Emerging Redox-Based Therapeutic Interventions

Emerging redox-based therapeutic interventions are at the forefront of innovative strategies aimed at restoring redox balance and mitigating chronic inflammatory diseases. These novel approaches encompass advanced nanomedicine, gene-based therapies, and the modulation of cell death pathways, each offering unique mechanisms to address OS and inflammation.

Nanotechnology has enabled the development of redox-responsive nanomedicines designed for targeted drug delivery and controlled release [175]. These nanosystems can respond to the oxidative environment characteristic of inflamed tissues, ensuring that therapeutic agents are released specifically at sites of inflammation [205]. Such targeted delivery enhances treatment efficacy while minimizing systemic side effects. For instance, ROS-responsive nanoparticles have been engineered to release anti-inflammatory drugs upon encountering high ROS levels in inflamed tissues [206].

Gene therapy offers a promising avenue for redox modulation by targeting specific genes involved in OS and inflammatory pathways. Techniques such as small interfering RNA and adenovirus-based gene delivery have been employed to modulate the expression of redox-regulating enzymes [176,207]. These interventions aim to enhance the expression of antioxidant enzymes or suppress pro-oxidant factors, thereby restoring redox homeostasis and alleviating inflammation. Recent studies have demonstrated the feasibility of these gene-based approaches in preclinical models, suggesting potential for clinical translation [166].

However, most current strategies focus on counteracting OS, with limited attention to the detrimental effects of RS. Novel interventions, such as selective inhibitors of excessive NADPH production or the controlled modulation of mitochondrial electron transport chain activity, are now being explored to mitigate RS in inflammatory and cancerous conditions [127].

In summary, these emerging redox-based therapeutic interventions represent a significant advancement in the management of chronic inflammatory diseases. By leveraging nanotechnology, gene therapy, and personalized medicine, these strategies offer targeted and efficient means to restore redox balance and mitigate inflammation. Ongoing research and clinical trials will be crucial in translating these innovative approaches into effective treatments for patients suffering from chronic inflammatory conditions. Future redox-based therapies should include biomarkers that assess both oxidative and RS, ensuring a fine-tuned therapeutic approach.

## 7. Conclusions

Redox balance is essential for immune homeostasis and inflammation control. While OS has been widely studied for its role in chronic inflammation, this review highlights the equally important, yet often underappreciated, contribution of RS. Excessive antioxidant activity and the accumulation of reducing equivalents can disrupt redox-sensitive signaling, impair immune responses, and contribute to disease progression. Both oxidative and reductive imbalances are implicated in autoimmune, cardiovascular, and neuroinflammatory conditions.

In contrast to previous reviews that have treated OS and RS as separate entities, our analysis emphasizes their reciprocal dynamics and shared regulatory mechanisms. We propose that chronic inflammatory diseases arise not from a unidirectional redox shift, but from a breakdown in the feedback systems that normally maintain redox equilibrium. This framework underscores the importance of evaluating redox imbalance as a spectrum rather than as a binary state.

Therapeutic strategies must therefore evolve beyond conventional antioxidant supplementation. Current approaches that aim solely to suppress ROS may inadvertently promote RS if not properly balanced. More refined interventions—such as selective modulation of Nrf2 and NF-κB, redox-sensitive enzyme targeting, and mitochondria-focused therapies—offer promising potential. Furthermore, integrating redox metabolomics and developing robust biomarkers capable of distinguishing oxidative from reductive states will be essential for advancing personalized treatments. Future research should focus on defining quantitative thresholds that distinguish physiological from pathological OS and RS, mapping disease-specific redox profiles, and exploring the interplay between redox status and immune regulation in greater depth. Particular attention should be given to the risks associated with excessive antioxidant intake in otherwise healthy individuals, as recent studies suggest this may contribute to RS and inflammation. Ultimately, restoring physiological redox signaling—rather than simply counteracting oxidants—should be guiding principle for next-generation therapies targeting redox imbalance in chronic inflammatory diseases.

## Figures and Tables

**Figure 1 antioxidants-14-00656-f001:**
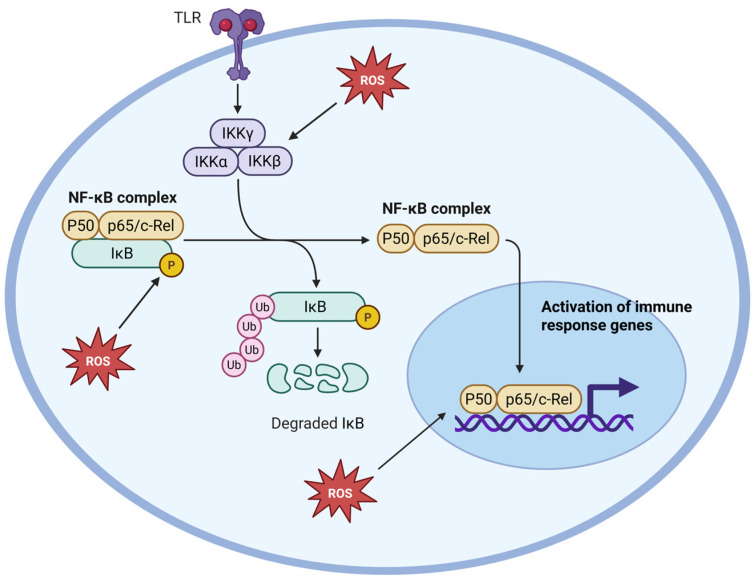
Reactive oxygen species (ROS)-mediated activation of the NF-κB signaling pathway. ROS, either produced endogenously or induced by extracellular stimuli, promote the activation of the IκB kinase (IKK) complex (IKKα, IKKβ, and IKKγ). Activated IKK phosphorylates IκB, the inhibitory subunit of the cytoplasmic NF-κB complex (p50/p65 or p50/c-Rel). ROS has also been shown to activate NF-κB through alternative IκB phosphorylation. Phosphorylated IκB undergoes ubiquitination and subsequent proteasomal degradation, liberating the NF-κB dimer. The released NF-κB translocates into the nucleus, where it binds DNA and activates the transcription of pro-inflammatory and immune response genes. ROS also modulate the DNA binding of NF-κB and amplify transcriptional activity, creating a feed-forward loop in inflammation. Created in https://BioRender.com (accessed on 15 May 2025).

**Figure 2 antioxidants-14-00656-f002:**
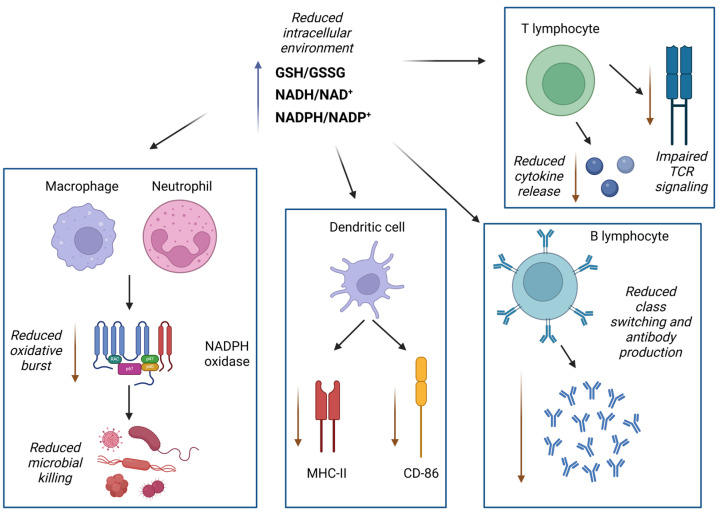
Reductive stress impairs immune cell function and promotes immune dysregulation. An excessively reduced intracellular environment—characterized by elevated GSH/GSSG, NADH/NAD^+^, and NADPH/NADP^+^ ratios—can compromise both innate and adaptive immunity. In macrophages and neutrophils, reductive stress suppresses NADPH oxidase activity, impairing the oxidative burst and microbial killing. Dendritic cells show reduced major histocompatibility complex (MHC)-II and CD86 expression, weakening T cell priming. In T lymphocytes, low ROS availability impairs T cell receptor (TCR) signaling and cytokine release, while B cells exhibit diminished antibody class switching and production. These immunosuppressive effects contribute to increased susceptibility to infections and chronic inflammation. Created in https://BioRender.com (accessed on 15 May 2025).

**Figure 3 antioxidants-14-00656-f003:**
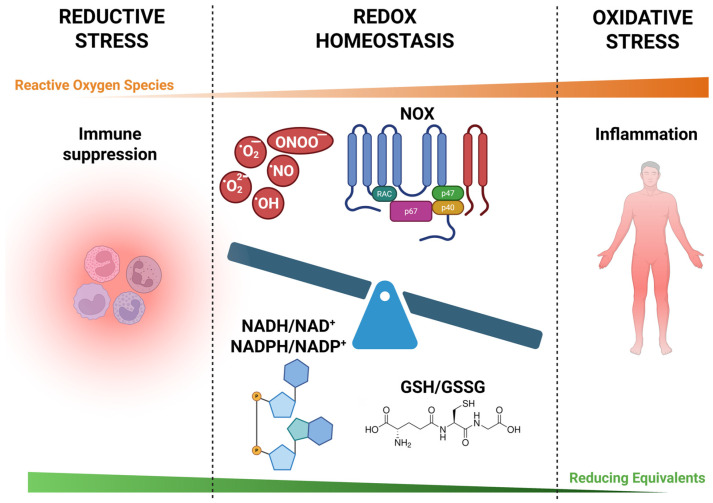
Redox regulation of immune responses: the continuum between reductive stress, redox homeostasis, and oxidative stress. Under physiological conditions (central panel), a balanced redox environment supports immune function by maintaining appropriate levels of reactive oxygen and nitrogen species (ROS/RNS), such as superoxide (^•^O_2_^−^), hydroxyl radical (^•^OH), nitric oxide (^•^NO), and peroxynitrite (ONOO^−^), primarily generated via regulated NOX activity. These species act as signaling molecules in immune activation, microbial killing, and tissue repair. In oxidative stress (right panel), excess ROS/RNS (red color) drive chronic inflammation and tissue injury, contributing to pathologies such as cardiovascular and neuroinflammatory diseases. In contrast, reductive stress (left panel), marked by increased intracellular reducing equivalents (e.g., high NADH/NAD^+^, NADPH/NADP^+^, and GSH/GSSG ratios, green color), leads to suppressed immune activation, impaired pathogen clearance, and dysregulated signaling. This overly reduced state may favor immune tolerance, impair cytokine responses, and promote metabolic dysfunction. The figure highlights the importance of maintaining redox homeostasis for immune integrity and inflammation control. Created in https://BioRender.com (accessed on 25 May 2025).

**Table 1 antioxidants-14-00656-t001:** Key markers of oxidative vs. reductive stress [95,96,97,98].

Category	Oxidative Stress Markers	Reductive Stress Markers
Reactive species	Superoxide (O_2_^•−^) Hydroxyl radicals (^•^OH)	Excess NADH and NADPH
Oxidation to cell membrane lipids	HNE MDA	Altered lipid peroxidation ratio
Protein damage	Carbonylated proteins Oxidized thiols	Increased free thiol groups and disrupted disulfide bonds
DNA damage	8-OHdG Nitrosylated DNA	Reduced DNA oxidation markers
Antioxidant levels	Decreased SOD, catalase, and glutathione	Overactive glutathione system

Abbreviations: HNE, hydroxynonenal; MDA, malondialdehyde; 8-OHdG, 8-hydroxy-2-deoxyguanosine; and SOD, superoxide dismutase.

**Table 2 antioxidants-14-00656-t002:** Comparison of oxidative and reductive stress in chronic diseases.

Disease	Oxidative Stress Role	Reductive Stress Role
Rheumatoid arthritis	Increased ROS leads to joint inflammation [120]	Reduced immune activation affects response [121]
Atherosclerosis	LDL oxidation triggers plaque formation [122]	Excess NADPH alters cholesterol metabolism [123]
Neurodegeneration	ROS damage neurons (Parkinson’s, Alzheimer’s) [87,124]	Excess reducing equivalents alters synaptic plasticity [125]
Diabetes	Mitochondrial ROS disrupt insulin signaling [126]	Increased NADH contributes to insulin resistance [70]

Abbreviations: ROS, reactive oxygen species; and LDLs, low-density lipoproteins.

**Table 3 antioxidants-14-00656-t003:** Current and emerging redox-based therapeutic strategies.

Therapeutic Category	Examples	Mechanism of Action	Targeted Redox Imbalance
Antioxidants [172]	Vitamin C, NAC, and resveratrol	ROS scavenging and NF-κB inhibition	Oxidative stress
Pharmacological agents [31,173]	Nrf2 activators NOX inhibitors	Modulate redox-sensitive transcription factors	Both
Dietary approaches [160,174]	Mediterranean diet Polyphenols	Enhancing endogenous antioxidant defenses	Both
Nanomedicine [175]	ROS-responsive nanoparticles	Targeted drug release in inflamed tissues	Oxidative stress
Gene therapy [176]	siRNA against NOX enzymes	Downregulating excess ROS production	Oxidative stress

Abbreviations: NAC, N-acetylcysteine; ROS, reactive oxygen species; NF-κB, nuclear factor-kappa B; and NOX, NADPH oxidases.

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
