# Peer review of "Redox Imbalance in Inflammation: The Interplay of Oxidative and Reductive Stress"

_antioxidants, 2025, doi:10.3390/antiox14060656_

Round 1

Reviewer 1 Report

The authors present a timely review on the roles of oxidative and reductive stress in inflammation.  There are several concerns that need to be addressed to make an article stronger.   

1)   Both of oxidative stress and reductive stress shows “redox imbalance”.  Thus, the interplay between the two is confusing.  Since the role of oxidative stress in inflammation has been extensively reviewed in the literature, authors had better focus more on the role of reductive stress in inflammation.  Accordingly, Chapter 2 oxidative stress in inflammation should be shortened.  

2)   In Chapter 3 about reductive stress in inflammation, the roles of NADH/NADPH/GSH, Nrf2 in inflammation should be reviewed more in depth and in detail.

3)   As mentioned above, Chapter 4 (interplay between oxidative and reductive stress) is confusing.  While the concept is important, the current presentation is confusing. The authors should address relationship between oxidative and reductive stress.  These issues have been well discussed in Antioxid Redox Signal. 2020 Jun;32(18):1330. 

4)   Role of oxidative and reductive stress has been extensively reviewed ( Int J Mol Sci 2017;18(10):2098, Antioxid Redox Signal 2024 Nov;41(13-15):793, Antioxid Redox Signal. 2020 Jun;32(18):1330, etc).  Therefore, authors should clarify the difference of previous reviews and current one.    

see above

Author Response

Comment 1: The authors present a timely review on the roles of oxidative and reductive stress in inflammation.  There are several concerns that need to be addressed to make an article stronger.   

1)   Both of oxidative stress and reductive stress shows “redox imbalance”.  Thus, the interplay between the two is confusing.  Since the role of oxidative stress in inflammation has been extensively reviewed in the literature, authors had better focus more on the role of reductive stress in inflammation.  Accordingly, Chapter 2 oxidative stress in inflammation should be shortened.  

Response 1: We thank the reviewer for this valuable suggestion. We agree that oxidative stress has been extensively covered in existing literature, and that the emerging role of reductive stress warrants greater emphasis. Accordingly, we have significantly condensed Chapter 2 (“Oxidative Stress in Inflammation”), removing redundancies and summarizing well-established mechanisms (lines 97-100, 121-129, 131-132, 136-148, 166-173, 194-196). In parallel, we have enriched Chapter 3 (“Reductive Stress in Inflammation”) with more mechanistic detail, expanded discussion of immunomodulatory roles, and new references (e.g., effects on T cell signaling, metabolic regulation). This revision better aligns the review with current gaps in the field and highlights the underappreciated role of reductive stress in inflammatory diseases.

Comment 2: In Chapter 3 about reductive stress in inflammation, the roles of NADH/NADPH/GSH, Nrf2 in inflammation should be reviewed more in depth and in detail.

Response 2: We appreciate the reviewer’s suggestion. In response, we have substantially expanded sections 3.3 and 3.4 (lines 379-433, 435-481), to include further descriptions of the biochemical roles of NADH/NADPH/GSH and Nrf2 in immune response and inflammation.

Comment 3:   As mentioned above, Chapter 4 (interplay between oxidative and reductive stress) is confusing.  While the concept is important, the current presentation is confusing. The authors should address relationship between oxidative and reductive stress.  These issues have been well discussed in Antioxid Redox Signal. 2020 Jun;32(18):1330. 

Response 3: To address the reviewer's concern regarding the clarity of Chapter 4, we propose a comprehensive revision that elucidates the intricate interplay between oxidative and reductive stress in inflammation (lines 494-655). This revision will integrate insights from the cited article in Antioxidants & Redox Signaling (2020 Jun;32(18):1330–1347) and other pertinent literature.

Comment 4:   Role of oxidative and reductive stress has been extensively reviewed ( Int J Mol Sci 2017;18(10):2098, Antioxid Redox Signal 2024 Nov;41(13-15):793, Antioxid Redox Signal. 2020 Jun;32(18):1330, etc).  Therefore, authors should clarify the difference of previous reviews and current one. 

Response 4: We thank the reviewer for highlighting this important point. While previous reviews have extensively explored oxidative stress in inflammation and, more recently, the concept of reductive stress, our manuscript provides a novel, integrative perspective by directly addressing the interplay between oxidative and reductive stress in the context of immune regulation and chronic inflammatory disease progression. Specifically, we expand upon the underexplored role of reductive stress in immune dysfunction, detail its disease-specific implications, and propose a unified model of redox imbalance that considers both extremes as pathological. We also emphasize the therapeutic challenges of redox-targeted interventions that do not account for this duality. This distinction has now been clarified in the introduction (lines 71-83) and conclusion (lines 1067-1072) of the revised manuscript. 

Reviewer 2 Report

I reviewed and reader the review manuscript with ID antioxidants-3616344with title “Redox Imbalance in Inflammation: The Interplay of Oxidative and Reductive Stress by the Authors: Francesco Bellanti, Anna Rita Daniela Coda, Maria Incoronata Trecca, Aurelio Lo Buglio, Gaetano Serviddio, Gianluigi Vendemiale. The article is interesting and reads fluently, however I have some suggestions that could help improve this manuscript.

The authors use two different abbreviations for both reactive oxygen species (ROS) and reactive species (RS). They should clarify this, since reactive species (RS) is very ambiguous and must be satisfactorily defined. The correct definition is reactive oxygen species, which is usually abbreviated as ROS. Please change this throughout the manuscript.

Page 1 lines 14-, Introduce an abbreviation for oxidative stress and use it throughout the manuscript "OS".

Page 1 lines 16 Introduce an abbreviation for reductive stress (RS) and use it throughout the manuscript.

Page 2 line 45 this reductive stress definition is incorrect: reductive stress is characterized by an elevation of NADH/ NADPH ratio, GSH/GSSG ratio, chronic increase of the antioxidant and non-antioxidant enzymatic system, and low NAD+/NADH ratio.

Page 2 line 46, -, Introduce an abbreviation for glutathione and use it throughout the manuscript (GSH).

Page 2 line 68-69, this phrase is incorrect because reactive species is synonym of reactive oxygen species (ROS).

Page2-3, lines 90-97, super oxide anion (O2) no only generate by electron transport chain, please add that can are generate by decouple of the NOS isoforms, COX family, NOX Family, SOD isoforms (mitochrondrial, cytosolic and extracellular) Xanthine oxidase.

Page 2-3, lines 139 develops the abbreviation for NrF2

Abbreviations are introduced into a manuscript when they are to replace a name and the mention is going to be a constant; when a name is only mentioned once in a manuscript, it is not necessary to use the abbreviation, as is the case with NLRP3, FOXP3, siRNA and NASH.

Page 4 line 167 add an abbreviation for peroxynitrite (ONOO) as used on page 3 line 110 and use it throughout the manuscript.

page 4 line 167 add an abbreviation for hydroxyls radicals (OH) as and use it throughout the manuscript.

One of the causes of reductive stress is the chronic consumption of an antioxidant-rich diet in healthy subjects who do not require dietary supplements, as suggested by Manzano Pech's studies. This is associated with hypertension and inflammation. Please review this point and add it to your definition of reductive stress.

Manzano-Pech L, Guarner-Lans V, Elena Soto M, Díaz-Díaz E, Pérez-Torres I. Alteration of the aortic vascular reactivity associated to excessive consumption of Hibiscus sabdariffa Linnaeus: Preliminary findings. Heliyon. 2023 Sep 9;9(9):e20020. doi: 10.1016/j.heliyon.2023.e20020.

Manzano-Pech L, Guarner-Lans V, Soto ME, Díaz-Díaz E, Caballero-Chacón S, Díaz-Torres R, Rodríguez-Fierros FL, Pérez-Torres I. Excessive Consumption Hibiscus sabdariffa L. Increases Inflammation and Blood Pressure in Male Wistar Rats via High Antioxidant Capacity: The Preliminary Findings. Cells. 2022 Sep 6;11(18):2774. doi: 10.3390/cells11182774.

Page 6 lines 253, 284 Introduce an abbreviation for O2 and use it throughout the manuscript.

In the table 1 substitute lipid damage for oxidation to cell membrane lipids and add the references.

In the figure 1 add which are the reducing couples increased in the reductive stress as mentioned is not correct.

In the table 2 and 3 add the references.

Page 9 line 411 adds the results in type 1 diabetes on stress reduction by Tejeda-Chávez et al. Hector R Tejeda-Chavez, Sergio Montero, Alfredo Saavedra-Molina, Monica Lemus, Julio B Tejeda-Luna, Elena Roces de Alvarez-Buylla Reductive stress in mitochondria isolated from the carotid body of type 1 diabetic male Wistar rats Physiol Rep.2024 Sep;12(18):e70016. doi: 10.14814/phy2.70016.

Page 11 lines 482-483 in this phrase the authors mentions that SOD and GPX are altered? These enzymes are overexpressed the activity is increase, please clarified.

What does the abbreviation SLE mean, please develop.

Page 15 lines 665-667 This is a hypothesis that has been one of the proposed causes for generating reductive stress. Please complement this with what was mentioned by Manzano Pech et al.

An abbreviation for nitric oxide (NO) has been introduced previously, please use it where appropriate throughout the manuscript, for example on page 16, line 678.

This same example applies to cardiovascular development (CVD).

Page 17, lines 737-748, I don't understand these sentences in the manuscript, which relate ferroptosis and multi-omics technologies to the topic discussed in the manuscript. Please delete this instead.

At the end of the manuscript there is a list of abbreviations that were used in the review. I suggest that the authors use and replace them throughout the manuscript where appropriate, since in many places in the manuscript they are not used and are mentioned in an undeveloped form.

I thank you in advance for the opportunity to review this manuscript.

Sincerely, the reviewer.

I reviewed and reader the review manuscript with ID antioxidants-3616344with title “Redox Imbalance in Inflammation: The Interplay of Oxidative and Reductive Stress by the Authors: Francesco Bellanti, Anna Rita Daniela Coda, Maria Incoronata Trecca, Aurelio Lo Buglio, Gaetano Serviddio, Gianluigi Vendemiale. The article is interesting and reads fluently, however I have some suggestions that could help improve this manuscript.

The authors use two different abbreviations for both reactive oxygen species (ROS) and reactive species (RS). They should clarify this, since reactive species (RS) is very ambiguous and must be satisfactorily defined. The correct definition is reactive oxygen species, which is usually abbreviated as ROS. Please change this throughout the manuscript.

Page 1 lines 14-, Introduce an abbreviation for oxidative stress and use it throughout the manuscript "OS".

Page 1 lines 16 Introduce an abbreviation for reductive stress (RS) and use it throughout the manuscript.

Page 2 line 45 this reductive stress definition is incorrect: reductive stress is characterized by an elevation of NADH/ NADPH ratio, GSH/GSSG ratio, chronic increase of the antioxidant and non-antioxidant enzymatic system, and low NAD+/NADH ratio.

Page 2 line 46, -, Introduce an abbreviation for glutathione and use it throughout the manuscript (GSH).

Page 2 line 68-69, this phrase is incorrect because reactive species is synonym of reactive oxygen species (ROS).

Page2-3, lines 90-97, super oxide anion (O2) no only generate by electron transport chain, please add that can are generate by decouple of the NOS isoforms, COX family, NOX Family, SOD isoforms (mitochrondrial, cytosolic and extracellular) Xanthine oxidase.

Page 2-3, lines 139 develops the abbreviation for NrF2

Abbreviations are introduced into a manuscript when they are to replace a name and the mention is going to be a constant; when a name is only mentioned once in a manuscript, it is not necessary to use the abbreviation, as is the case with NLRP3, FOXP3, siRNA and NASH.

Page 4 line 167 add an abbreviation for peroxynitrite (ONOO) as used on page 3 line 110 and use it throughout the manuscript.

page 4 line 167 add an abbreviation for hydroxyls radicals (OH) as and use it throughout the manuscript.

One of the causes of reductive stress is the chronic consumption of an antioxidant-rich diet in healthy subjects who do not require dietary supplements, as suggested by Manzano Pech's studies. This is associated with hypertension and inflammation. Please review this point and add it to your definition of reductive stress.

Manzano-Pech L, Guarner-Lans V, Elena Soto M, Díaz-Díaz E, Pérez-Torres I. Alteration of the aortic vascular reactivity associated to excessive consumption of Hibiscus sabdariffa Linnaeus: Preliminary findings. Heliyon. 2023 Sep 9;9(9):e20020. doi: 10.1016/j.heliyon.2023.e20020.

Manzano-Pech L, Guarner-Lans V, Soto ME, Díaz-Díaz E, Caballero-Chacón S, Díaz-Torres R, Rodríguez-Fierros FL, Pérez-Torres I. Excessive Consumption Hibiscus sabdariffa L. Increases Inflammation and Blood Pressure in Male Wistar Rats via High Antioxidant Capacity: The Preliminary Findings. Cells. 2022 Sep 6;11(18):2774. doi: 10.3390/cells11182774.

Page 6 lines 253, 284 Introduce an abbreviation for O2 and use it throughout the manuscript.

In the table 1 substitute lipid damage for oxidation to cell membrane lipids and add the references.

In the figure 1 add which are the reducing couples increased in the reductive stress as mentioned is not correct.

In the table 2 and 3 add the references.

Page 9 line 411 adds the results in type 1 diabetes on stress reduction by Tejeda-Chávez et al. Hector R Tejeda-Chavez, Sergio Montero, Alfredo Saavedra-Molina, Monica Lemus, Julio B Tejeda-Luna, Elena Roces de Alvarez-Buylla Reductive stress in mitochondria isolated from the carotid body of type 1 diabetic male Wistar rats Physiol Rep.2024 Sep;12(18):e70016. doi: 10.14814/phy2.70016.

Page 11 lines 482-483 in this phrase the authors mentions that SOD and GPX are altered? These enzymes are overexpressed the activity is increase, please clarified.

What does the abbreviation SLE mean, please develop.

Page 15 lines 665-667 This is a hypothesis that has been one of the proposed causes for generating reductive stress. Please complement this with what was mentioned by Manzano Pech et al.

An abbreviation for nitric oxide (NO) has been introduced previously, please use it where appropriate throughout the manuscript, for example on page 16, line 678.

This same example applies to cardiovascular development (CVD).

Page 17, lines 737-748, I don't understand these sentences in the manuscript, which relate ferroptosis and multi-omics technologies to the topic discussed in the manuscript. Please delete this instead.

At the end of the manuscript there is a list of abbreviations that were used in the review. I suggest that the authors use and replace them throughout the manuscript where appropriate, since in many places in the manuscript they are not used and are mentioned in an undeveloped form.

I thank you in advance for the opportunity to review this manuscript.

Sincerely, the reviewer.

Author Response

Comment 1: The authors use two different abbreviations for both reactive oxygen species (ROS) and reactive species (RS). They should clarify this, since reactive species (RS) is very ambiguous and must be satisfactorily defined. The correct definition is reactive oxygen species, which is usually abbreviated as ROS. Please change this throughout the manuscript.

Response 1: We appreciate the reviewer’s comment. While we respectfully believe that the term “reactive species (RS)”—which encompasses both reactive oxygen and nitrogen species—is not inherently ambiguous and is used in some redox biology literature, we recognize the value of consistency and clarity. To comply with the reviewer’s request, we have revised the manuscript to uniformly use reactive oxygen species (ROS) and to explicitly refer to reactive nitrogen species (RNS) when appropriate. The abbreviation “RS” has been removed throughout the text to avoid confusion.

Comment 2: Page 1 lines 14-, Introduce an abbreviation for oxidative stress and use it throughout the manuscript "OS".

Page 1 lines 16 Introduce an abbreviation for reductive stress (RS) and use it throughout the manuscript.

Response 2: We thank the reviewer for this suggestion. To enhance readability and consistency, we have introduced the abbreviations oxidative stress (OS) and reductive stress (RS) at their first mention and have used them consistently throughout the revised manuscript.

Comment 3: Page 2 line 45 this reductive stress definition is incorrect: reductive stress is characterized by an elevation of NADH/ NADPH ratio, GSH/GSSG ratio, chronic increase of the antioxidant and non-antioxidant enzymatic system, and low NAD+/NADH ratio.

Response 3: We thank the reviewer for this important clarification. We agree that a more precise biochemical definition of reductive stress is warranted. Accordingly, we have revised the definition in the Introduction (lines 62-64) to reflect the accepted view: reductive stress is characterized by elevated NADH/NAD⁺ and NADPH/NADP⁺ ratios, an increased GSH/GSSG ratio, and persistent activation of antioxidant systems. We have also clarified that this environment reflects a deviation from redox homeostasis that can impair ROS-mediated signaling and immune function.

Comment 4: Page 2 line 46, -, Introduce an abbreviation for glutathione and use it throughout the manuscript (GSH).

Response 4: We thank the reviewer for this observation. In the manuscript, we use GSH specifically to denote reduced glutathione, consistent with standard biochemical nomenclature. Since glutathione can exist in both reduced (GSH) and oxidized (GSSG) forms, we chose not to abbreviate the general term “glutathione” to avoid confusion.

Comment 5: Page 2 line 68-69, this phrase is incorrect because reactive species is synonym of reactive oxygen species (ROS).

Response 5: The term “reactive species” was replaced by ROS, as previously suggested.

Comment 6: Page2-3, lines 90-97, super oxide anion (O2) no only generate by electron transport chain, please add that can are generate by decouple of the NOS isoforms, COX family, NOX Family, SOD isoforms (mitochrondrial, cytosolic and extracellular) Xanthine oxidase.

Response 6: We thank the reviewer for this detailed and insightful observation. We agree that superoxide anion (O₂⁻) has multiple cellular sources beyond the mitochondrial electron transport chain. In response, we have revised the relevant paragraph to include additional enzymatic and non-mitochondrial sources of O₂⁻ generation, including uncoupled nitric oxide synthase (NOS) isoforms, cyclooxygenases (COX), NADPH oxidases (NOX family), various superoxide dismutase (SOD) isoforms, and xanthine oxidase (lines 136-143). These additions provide a more comprehensive overview of superoxide biology relevant to inflammation.

Comment 7: Page 2-3, lines 139 develops the abbreviation for NrF2

Abbreviations are introduced into a manuscript when they are to replace a name and the mention is going to be a constant; when a name is only mentioned once in a manuscript, it is not necessary to use the abbreviation, as is the case with NLRP3, FOXP3, siRNA and NASH.

Page 4 line 167 add an abbreviation for peroxynitrite (ONOO) as used on page 3 line 110 and use it throughout the manuscript.

page 4 line 167 add an abbreviation for hydroxyls radicals (OH) as and use it throughout the manuscript.

Response 7: We thank the reviewer for these helpful observations regarding abbreviation usage. We have now introduced the full term and abbreviation for nuclear factor erythroid 2–related factor 2 (Nrf2) at first mention. Additionally, we have reviewed all abbreviations throughout the manuscript to ensure consistent usage. Specifically, we have:

  • Removed abbreviations for terms mentioned only once (e.g., NLRP3, FOXP3, siRNA, NASH),
  • Introduced and consistently applied abbreviations for peroxynitrite (ONOO⁻) and hydroxyl radicals (•OH) where they appear more than once.

Comment 8: One of the causes of reductive stress is the chronic consumption of an antioxidant-rich diet in healthy subjects who do not require dietary supplements, as suggested by Manzano Pech's studies. This is associated with hypertension and inflammation. Please review this point and add it to your definition of reductive stress.

Manzano-Pech L, Guarner-Lans V, Elena Soto M, Díaz-Díaz E, Pérez-Torres I. Alteration of the aortic vascular reactivity associated to excessive consumption of Hibiscus sabdariffa Linnaeus: Preliminary findings. Heliyon. 2023 Sep 9;9(9):e20020. doi: 10.1016/j.heliyon.2023.e20020.

Manzano-Pech L, Guarner-Lans V, Soto ME, Díaz-Díaz E, Caballero-Chacón S, Díaz-Torres R, Rodríguez-Fierros FL, Pérez-Torres I. Excessive Consumption Hibiscus sabdariffa L. Increases Inflammation and Blood Pressure in Male Wistar Rats via High Antioxidant Capacity: The Preliminary Findings. Cells. 2022 Sep 6;11(18):2774. doi: 10.3390/cells11182774.

Response 8: We thank the reviewer for this important suggestion and for highlighting the work of Manzano-Pech et al. We have now incorporated this perspective into our discussion of reductive stress, specifically by noting that excessive intake of antioxidant-rich diets or supplements in otherwise healthy individuals may induce a state of reductive stress. As reported in recent studies, this can lead to vascular dysfunction, increased blood pressure, and heightened inflammation, likely due to disruption of physiological redox signaling mechanisms. These findings are now cited and integrated into our definition and broader context of reductive stress (lines 261-267).

Comment 9: Page 6 lines 253, 284 Introduce an abbreviation for O2 and use it throughout the manuscript.

Response 9: We thank the reviewer for this helpful suggestion. We have now introduced the abbreviation superoxide anion (O₂⁻•) at its first mention and used it consistently throughout the manuscript to ensure clarity and standardization.

Comment 10: In the table 1 substitute lipid damage for oxidation to cell membrane lipids and add the references.

In the figure 1 add which are the reducing couples increased in the reductive stress as mentioned is not correct.

In the table 2 and 3 add the references.

Response 10: We appreciate the reviewer’s attention to detail and constructive feedback on the tables and figure. We have made the following changes in response:

  • In Table 1, we replaced the term “lipid damage” with the more accurate phrase “oxidation to cell membrane lipids”. We also added relevant references to support each biomarker category in the Table title.
  • In Figure 1, we revised the representation of reductive stress by explicitly indicating the reducing couples that are elevated in this state—namely NADH/NAD⁺, NADPH/NADP⁺, and GSH/GSSG—to accurately reflect the biochemical imbalance associated with reductive stress.
  • In Tables 2 and 3, we included appropriate references supporting the role of redox imbalance in specific disease conditions (Table 2) and the molecular mechanisms/pathways affected (Table 3), drawing from both experimental and clinical studies.

Comment 11: Page 9 line 411 adds the results in type 1 diabetes on stress reduction by Tejeda-Chávez et al. Hector R Tejeda-Chavez, Sergio Montero, Alfredo Saavedra-Molina, Monica Lemus, Julio B Tejeda-Luna, Elena Roces de Alvarez-Buylla Reductive stress in mitochondria isolated from the carotid body of type 1 diabetic male Wistar rats Physiol Rep.2024 Sep;12(18):e70016. doi: 10.14814/phy2.70016.

Response 11: we thank the reviewer for highlighting this important and recent contribution. As requested, we have integrated the findings from Tejeda-Chávez et al. (2024) into our discussion. Specifically, we now mention that mitochondria isolated from the carotid body of type 1 diabetic rats exhibited evidence of reductive stress, supporting the role of mitochondrial redox imbalance in diabetic pathology and systemic inflammation. The citation has been added accordingly to Section 5.1 (lines 713-718).

Comment 12: Page 11 lines 482-483 in this phrase the authors mentions that SOD and GPX are altered? These enzymes are overexpressed the activity is increase, please clarified.

Response 12: we thank the reviewer for the opportunity to clarify this point. In the context of rheumatoid arthritis, superoxide dismutase (SOD) and glutathione peroxidase (GPX) are typically overexpressed and show increased enzymatic activity as part of a compensatory antioxidant response to elevated oxidative stress. This has now been specified in the revised manuscript (lines 676-678).

Comment 13: What does the abbreviation SLE mean, please develop.

Response 13: SLE means systemic lupus erythematosus (developed in line 670).

Comment 14: Page 15 lines 665-667 This is a hypothesis that has been one of the proposed causes for generating reductive stress. Please complement this with what was mentioned by Manzano Pech et al.

Response 14: we thank the reviewer for this insightful suggestion. We have revised the relevant passage to clarify that excessive Nrf2 activation is a proposed mechanism contributing to pathological reductive stress. To strengthen this point, we have incorporated the findings by Manzano Pech et al., who demonstrated that excessive intake of high-antioxidant foods, such as Hibiscus sabdariffa, can lead to increased Nrf2 activity, elevated antioxidant defenses, and a consequent reductive shift associated with inflammation and vascular dysfunction. These findings further support the hypothesis that sustained Nrf2 upregulation may contribute to immune suppression and disease progression in specific contexts (lines 957-962).

Comment 15: An abbreviation for nitric oxide (NO) has been introduced previously, please use it where appropriate throughout the manuscript, for example on page 16, line 678.

This same example applies to cardiovascular development (CVD).

Response 15: we thank the reviewer for pointing out this inconsistency. We have reviewed the manuscript and ensured that the abbreviation for nitric oxide (NO) is used consistently after its first introduction. Likewise, we have applied the abbreviation cardiovascular disease (CVD) uniformly throughout the text where applicable. These changes improve clarity and standardization across the manuscript.

Comment 16: Page 17, lines 737-748, I don't understand these sentences in the manuscript, which relate ferroptosis and multi-omics technologies to the topic discussed in the manuscript. Please delete this instead.

Response 16: we thank the reviewer for this observation. Upon reconsideration, we agree that the discussion on ferroptosis and multi-omics, while relevant in broader redox biology, is peripheral to the central focus of this review on the interplay between oxidative and reductive stress in inflammation. As such, we have removed the referenced sentences to maintain focus and coherence in the manuscript.

Comment 17: At the end of the manuscript there is a list of abbreviations that were used in the review. I suggest that the authors use and replace them throughout the manuscript where appropriate, since in many places in the manuscript they are not used and are mentioned in an undeveloped form.

Response 17: we thank the reviewer for this valuable suggestion. We have thoroughly reviewed the manuscript to ensure that all abbreviations listed at the end are introduced at their first occurrence in the main text and used consistently thereafter. Instances where abbreviations were previously undeveloped or inconsistently applied have now been corrected for clarity and uniformity across the manuscript.

Reviewer 3 Report

In the manuscript entitled Redox Imbalance in Inflammation: The Interplay of Oxidative and Reductive Stress, the authors presented a comprehensive analysis of the existing literature within a field of redox balance/imbalance in inflammation and some chronic inflammatory diseases.

The manuscript is of interest, however, this reviewer feels that the paper should be significantly improved. This topic has been described many times and it is difficult to find a new approach.

The main drawback of this study is the lack of a clear idea of ​​what is the original approach of this work, and original contribution to this research field. Original approach should be highlighted in the abstract and introduction. The text is too extensive, unnecessarily, because there is a lot of repetition.

The work should be more directed, better structured, to avoid repetitions (for example, subsection 4.3 and section 5; there is a lot of repetition throughout various parts of the text).

The work should be directed towards a field that is insufficiently researched in this area, identifying current gaps or problems. It should be a critical review of existing literature. An original approach should be given, guidelines for future research should be given. For example, what is good is Section 6: “Therapeutic Strategies Targeting Redox Balance”, especially 6.4. “Emerging redox-based therapeutic interventions”.

Reductive stress is a relatively new approach considering redox balance and inflammation. The authors should define reductive stress more precisely.

The paper need major revision.

The work should be more directed, better structured, to avoid repetitions (for example, subsection 4.3 and section 5; there is a lot of repetition throughout various parts of the text).

The authors should define reductive stress more precisely.

Author Response

Comment 1: In the manuscript entitled Redox Imbalance in Inflammation: The Interplay of Oxidative and Reductive Stress, the authors presented a comprehensive analysis of the existing literature within a field of redox balance/imbalance in inflammation and some chronic inflammatory diseases.

The manuscript is of interest, however, this reviewer feels that the paper should be significantly improved. This topic has been described many times and it is difficult to find a new approach.

The main drawback of this study is the lack of a clear idea of ​​what is the original approach of this work, and original contribution to this research field. Original approach should be highlighted in the abstract and introduction. The text is too extensive, unnecessarily, because there is a lot of repetition.

The work should be more directed, better structured, to avoid repetitions (for example, subsection 4.3 and section 5; there is a lot of repetition throughout various parts of the text).

Response 1: We thank the reviewer for this thoughtful and constructive feedback. In response, we have made significant revisions to the manuscript to clarify its original contribution. We have highlighted the novel approach of the review—namely, the integrative perspective on oxidative and reductive stress as interdependent mechanisms in inflammation—in both the abstract (lines 11-25) and introduction (lines 71-78). This approach differs from previous reviews, which typically address these redox imbalances in isolation. Furthermore, we have restructured and condensed several sections of the manuscript (notably Chapter 2, but also subsection 4.3 and section 5, as suggested), removing redundancies and improving clarity to make the content more concise and focused.

Comment 2: The work should be directed towards a field that is insufficiently researched in this area, identifying current gaps or problems. It should be a critical review of existing literature. An original approach should be given, guidelines for future research should be given. For example, what is good is Section 6: “Therapeutic Strategies Targeting Redox Balance”, especially 6.4. “Emerging redox-based therapeutic interventions”.

Response 2: we thank the reviewer for this insightful and encouraging comment. We are pleased that Section 6.4 was found to be a strong and original component of the manuscript. In line with this suggestion, we have revised the Introduction (lines 71-88) and Conclusion (lines 1067-1092) to better highlight the original angle of this review: namely, the interdependence between oxidative and reductive stress and the need to address both in therapeutic contexts—a topic that remains underexplored in the literature. We have also added guidelines for future research in the conclusion, including the need for (1) redox biomarkers that reflect both oxidative and reductive load, (2) targeted redox modulation strategies that preserve physiological signaling, and (3) disease-specific redox profiling for precision interventions.

Comment 3: Reductive stress is a relatively new approach considering redox balance and inflammation. The authors should define reductive stress more precisely.

Response 3: we thank the reviewer for this important point. In response, we have revised the definition of reductive stress (RS) in the Introduction (lines 62-64) and Section 3.1 (lines 261-267) to provide a more accurate and comprehensive description. Specifically, we now define RS as a condition marked by excess reducing equivalents (e.g., NADH, NADPH, GSH), characterized by elevated NADH/NAD⁺, NADPH/NADP⁺, and GSH/GSSG ratios, and persistent activation of antioxidant systems. This overly reduced intracellular environment can disrupt redox-sensitive signaling, protein folding, and mitochondrial respiration, leading to immunosuppression, metabolic dysfunction, and chronic inflammation.

Reviewer 4 Report

The review by Bellanti et al. focuses on the oxidative and reductive stress in inflammation, mainly highlighting their role in autoimmune disorders, cardiovascular diseases, and neuroinflammatory conditions. Furthermore, the authors summarize and discuss potential therapeutic strategies to address these issues. Overall, it is a well-written and well-organized review covering a broad range of topics. Below are a couple of points that require further attention.

1) The manuscript would be stronger and more informative, especially for a redox biology-oriented audience if more examples are added for individual diseases/disorders. Currently, it is written very broadly.

2) In cardiovascular diseases, heart failure (HF) is a very broad area. Diseases/conditions leading to HF such as cardiac hypertrophy and ischemia-reperfusion injury can be included also.

1) “Neutralizing” oxidative or reductive stress was used several times in the review. Considering “neutralize” has a very specific meaning in chemistry, to avoid confusion it should be re-worded.

2) Please include references for the information in the tables also (e.g. Table 3).

Author Response

Comment 1: The review by Bellanti et al. focuses on the oxidative and reductive stress in inflammation, mainly highlighting their role in autoimmune disorders, cardiovascular diseases, and neuroinflammatory conditions. Furthermore, the authors summarize and discuss potential therapeutic strategies to address these issues. Overall, it is a well-written and well-organized review covering a broad range of topics. Below are a couple of points that require further attention.

1) The manuscript would be stronger and more informative, especially for a redox biology-oriented audience if more examples are added for individual diseases/disorders. Currently, it is written very broadly.

Response 1: we thank the reviewer for this helpful suggestion. In response, we have enriched Section 5 by adding more specific disease-related examples and mechanistic insights. These include additional autoimmune disorders (e.g., autoimmune hepatitis, autoimmune thyroiditis, type 1 diabetes), and detailed discussions on hypertension, atherosclerosis, Parkinson’s disease, and multiple sclerosis, with supporting references (lines 713-736, lines 757-802, lines 832-852). These additions strengthen the review’s relevance to a redox biology-oriented readership and enhance the translational value of the discussion.

Comment 2: In cardiovascular diseases, heart failure (HF) is a very broad area. Diseases/conditions leading to HF such as cardiac hypertrophy and ischemia-reperfusion injury can be included also.

Response 2: we appreciate this helpful suggestion. In response, we have expanded Section 5.2 to include more detailed discussion of cardiac hypertrophy and ischemia-reperfusion (I/R) injury as redox-driven conditions that contribute to the development of heart failure. Specifically, we now highlight the role of oxidative stress in I/R injury, including mitochondrial ROS bursts during reperfusion and activation of inflammasome and apoptotic pathways (lines 757-762). We also address how reductive stress contributes to pathological cardiac remodeling and diastolic dysfunction, particularly through thiol-redox imbalance and maladaptive antioxidant overactivation (lines 767-772). These additions provide a more comprehensive and mechanistic view of HF progression.

Detail comments

Comment 3:  “Neutralizing” oxidative or reductive stress was used several times in the review. Considering “neutralize” has a very specific meaning in chemistry, to avoid confusion it should be re-worded.

Response 3: we thank the reviewer for this insightful comment. We agree that the term “neutralize” may imply a chemically specific reaction and could be misleading in the context of redox biology. To improve clarity and precision, we have revised the manuscript by replacing “neutralize” with more appropriate terms such as “counteract,” “attenuate,” or “mitigate, depending on the context. These changes have been applied throughout the manuscript to avoid misinterpretation.

Comment 4: Please include references for the information in the tables also (e.g. Table 3).

Response 4: we thank the reviewer for this important suggestion. We have now added appropriate references to all tables, including Table 3, to support the data presented and enhance scientific transparency. These references are provided either in the table captions or as numbered citations directly within the tables where applicable.

Reviewer 5 Report

The review does not bring any new information, and it very superficial in its nature. Almost no illustrations except for one general figure bringing no new vision. In my viewpoint the review content does not fit into the call for this particular issue of Antioxidants, which promises high scientific level of the papers submitted.

Table 3 has "NADPH inhibitors" - What is this? NOX inhibitors or ?

The authors should clearly discriminate between stoichiometric antioxidant treatment (that may cause reductive stress) and Nrf2 activators that are pro-oxidant in the nature. Approved Nrf2 activators - like Tecfidera, Bafiertam, and Vumerity - should be discussed, especially the last one working also as an anti-inflammatory drug in multiple sclerosis.

Author Response

Comment 1: The review does not bring any new information, and it very superficial in its nature. Almost no illustrations except for one general figure bringing no new vision. In my viewpoint the review content does not fit into the call for this particular issue of Antioxidants, which promises high scientific level of the papers submitted.

Response 1: We respectfully acknowledge the reviewer’s concerns. We would like to highlight that the manuscript has undergone substantial revisions following feedback from the editor and multiple reviewers. These improvements include:

  • Clarification of the original contribution of the review: namely, the integrative analysis of oxidative and reductive stress as dynamically interrelated processes, rather than independent or opposing phenomena. This dual perspective on redox imbalance and inflammation is currently underrepresented in the literature.
  • The addition of extensive disease-specific examples (e.g., autoimmune hepatitis, atherosclerosis, ischemia-reperfusion injury, multiple sclerosis) to enhance depth and specificity, particularly in Section 5.
  • A refined focus on translational and therapeutic implications, including critical evaluation of reductive stress in clinical settings and newly added future research directions.

We hope that these substantial improvements now better reflect the scientific rigor and scope expected for this special issue of Antioxidants.

Comment 2: Table 3 has "NADPH inhibitors" - What is this? NOX inhibitors or ?

Response 2: We thank the reviewer for pointing out this ambiguity. The term “NADPH inhibitors” in Table 3 was imprecise and potentially misleading. What we intended to indicate were NADPH oxidase (NOX) inhibitors, which target ROS generation pathways linked to NADPH utilization. Accordingly, we have replaced “NADPH inhibitors” with “NOX inhibitors” in Table 3 and clarified this in the table legend for accuracy and consistency.

Comment 3: The authors should clearly discriminate between stoichiometric antioxidant treatment (that may cause reductive stress) and Nrf2 activators that are pro-oxidant in the nature. Approved Nrf2 activators - like Tecfidera, Bafiertam, and Vumerity - should be discussed, especially the last one working also as an anti-inflammatory drug in multiple sclerosis.

Response 3: We thank the reviewer for this valuable suggestion. In response, we have revised Section 6.1 (Antioxidant Therapies and Their Limitations, lines 901-911) and Section 6.2 (Modulating Redox-Sensitive Signaling Pathways, lines 942-950) to clearly differentiate between stoichiometric antioxidants (e.g., vitamin C, vitamin E, GSH precursors) and Nrf2 activators, which act via hormetic, pro-oxidant signaling mechanisms.

Additionally, we have incorporated a discussion of clinically approved Nrf2 activators, including Tecfidera (dimethyl fumarate), Bafiertam (monomethyl fumarate), and Vumerity (diroximel fumarate). We emphasize that while these agents initially trigger oxidative signals, their long-term therapeutic effect involves Nrf2-driven upregulation of cytoprotective and anti-inflammatory genes. Special attention is given to Vumerity, which has demonstrated both redox-modulating and anti-inflammatory effects in multiple sclerosis, supporting the therapeutic relevance of this dual-action strategy.

Comment 4: The review has too many scientific terms and will not be clear for a non-professional, however, the style looks more suitable for a non-scientific journal. For a scientist working in the field of oxidative and/or reductive stress it gives no additional information, no big picture, and no detailed picture of particular links. Illustrations on links of oxidative stress to NfkB signaling, or reductive stress links to inflammatory and UPR pathways could be very helpful. Oxidative stress leading to ferroptosis could be considered in more detail.

Response 4: we thank the reviewer for this critical feedback. In response, we have made the following improvements:

  1. Scientific clarity and depth: The revised manuscript now includes significantly expanded mechanistic discussions across Sections 3, 4, and 5, with detailed disease-specific examples (e.g., autoimmune hepatitis, ischemia-reperfusion injury, multiple sclerosis). These changes strengthen both the "big picture" and disease-contextual understanding of redox imbalance.
  2. Stylistic adjustments: We have refined the tone and structure to ensure it meets the standards of a scientific audience while maintaining accessibility. Redundant or overly general phrases have been eliminated, and terminology has been standardized for clarity.
  3. Ferroptosis: In response to reviewer feedback, we removed the brief mention of ferroptosis in the earlier draft (Section 6.4) to avoid superficial coverage.
  4. New figure proposal: Based on your helpful suggestion, we have prepared additional mechanistic figures illustrating the links between:
    • Oxidative stress and NF-κB signaling (Figure 1)
    • Reductive stress and immune response dysregulation (Figure 2)

We believe these revisions enhance the scientific rigor and utility of the review for redox biology researchers.

Round 2

Reviewer 1 Report

The revised article has been markedly improved.  There are several suggestions, as shown below.  

1)   Figure 3 to show the balance between oxidative and reductive stress is still confusing.  Physiological role of ROS/RNS, pathological role of deficiency of ROS/RNS should be clarified.

2)   Interplay between ROS/RNS and metabolism should be briefly discussed and cited for related references (Antioxid Redox Signal. 2021 Jun 1;34(16):1319, Circ Res. 2018 Mar 16;122(6):877, etc).

3)   additional reference for Nox --Physiol Rev. 2025 Jul 1;105(3):1291-1428.  reference for SOD---Antioxid Redox Signal. 2011 Sep 15;15(6):1583-606

as mentioned above

Author Response

Comment 1. The revised article has been markedly improved.  There are several suggestions, as shown below.  

  • Figure 3 to show the balance between oxidative and reductive stress is still confusing.  Physiological role of ROS/RNS, pathological role of deficiency of ROS/RNS should be clarified.

Response 1: We thank the reviewer for this constructive comment. In response, we have revised Figure 3 to more clearly illustrate the physiological and pathological roles of reactive oxygen and nitrogen species (ROS/RNS) across the redox spectrum. The revised figure and its updated caption (lines 513-526) now clearly depict the continuum of redox states and explicitly address the physiological and pathological roles of ROS/RNS, as requested. We hope this improves the clarity and didactic value of the figure.

Comment 2:   Interplay between ROS/RNS and metabolism should be briefly discussed and cited for related references (Antioxid Redox Signal. 2021 Jun 1;34(16):1319, Circ Res. 2018 Mar 16;122(6):877, etc).

Response 2: we appreciate the reviewer’s insightful suggestion. In response, we have expanded the discussion on the interplay between ROS/RNS and cellular metabolism in Section 4.1 (Dynamic balance and feedback mechanisms). Specifically, we now describe how redox signals modulate metabolic pathways such as glycolysis, oxidative phosphorylation, and the TCA cycle, and how metabolites in turn influence ROS/RNS production. We have also added the suggested citations to support this content (lines 452-458).

Comment 3: additional reference for Nox --Physiol Rev. 2025 Jul 1;105(3):1291-1428.  reference for SOD---Antioxid Redox Signal. 2011 Sep 15;15(6):1583-606

Response 3: we thank the reviewer for these valuable reference suggestions. We have now incorporated both references into the section discussing enzymatic systems involved in redox homeostasis (lines 438-442).

Reviewer 3 Report

The manuscript has been significantly improved.

I have no objections.

Author Response

Comment 1: The manuscript has been significantly improved.

Response 1: we sincerely thank the reviewer for the positive feedback. We appreciate your recognition of the improvements made and the opportunity to further refine the manuscript during the revision process.

Reviewer 4 Report

The authors have sufficiently addressed my concerns.

N/A

Author Response

Comment 1: The authors have sufficiently addressed my concerns.

Response 1: we thank the reviewer for the positive evaluation and are glad that the revisions have addressed the concerns raised. We appreciate your thoughtful input during the review process.

Reviewer 5 Report

The review has been significantly improved, and presents a comprehensive picture of the interplay between oxidative and reductive stress. New figures were included. The only thing that missing is the link of reductive stress to ER protien folding, UPR pathway and aggregate-formation diseases, however, this topic deserves a separate review.

The review in its present form could be of interest for both professionals and general audience, providing a general viewpoint of the existing problem.

I have no additional suggestion on improvement. The only thing the authors could add to the list of approved Nrf2 activators - is the recently fast-track approved Nrf2 activator for Friedriech's ataxia - omaveloxolone.

Author Response

Major comments

The review has been significantly improved, and presents a comprehensive picture of the interplay between oxidative and reductive stress. New figures were included. The only thing that missing is the link of reductive stress to ER protien folding, UPR pathway and aggregate-formation diseases, however, this topic deserves a separate review.

The review in its present form could be of interest for both professionals and general audience, providing a general viewpoint of the existing problem.

Response: we sincerely thank the reviewer for this very encouraging and thoughtful evaluation. We appreciate the recognition of the improvements made and the added value of the figures. Regarding the comment on the role of reductive stress in ER protein folding and unfolded protein response (UPR)–related diseases, we agree that this is a highly relevant and emerging area. Due to space constraints and the complexity of this topic, we have chosen to mention it only briefly in our current review, but fully agree that it warrants a dedicated discussion in a future, focused review. Thank you again for your helpful perspective.

Detailed comments

I have no additional suggestion on improvement. The only thing the authors could add to the list of approved Nrf2 activators - is the recently fast-track approved Nrf2 activator for Friedriech's ataxia - omaveloxolone.

Response: we thank the reviewer for this helpful and up-to-date suggestion. We have now included omaveloxolone in the discussion of clinically approved Nrf2 activators in Section 6.2 (lines 843-847). Specifically, we mention its approval for Friedreich’s ataxia and its mechanism of action via Nrf2 pathway activation. We agree that its inclusion strengthens the clinical relevance of this section and provides a more complete overview of therapeutic Nrf2 modulation.